

# Fragmentation-induced localization and boundary charges in dimensions two and above

Julius Lehmann[1*], Pablo Sala de Torres-Solanot[1*],
Frank Pollmann[1,2] and Tibor Rakovszky[3]

**1** Department of Physics, TFK, Technische Universität München,
James-Franck-Straße 1, D-85748 Garching, Germany
**2** Munich Center for Quantum Science and Technology (MCQST),
Schellingstr. 4, D-80799 München, Germany
**3** Department of Physics, Stanford University, Stanford, CA 94305, USA

⋆ These authors contributed equally to this work

## Abstract

We study higher dimensional models with symmetric correlated hoppings, which generalize a one-dimensional model introduced in the context of dipole-conserving dynamics. We prove rigorously that whenever the local configuration space takes its smallest non-trivial value, these models exhibit localized behavior due to fragmentation, in any dimension. For the same class of models, we then construct a hierarchy of conserved quantities that are power-law localized at the boundary of the system with increasing powers. Combining these with Mazur's bound, we prove that boundary correlations are infinitely long lived, even when the bulk is not localized. We use our results to construct quantum Hamiltonians that exhibit the analogues of strong zero modes in two and higher dimensions.



# 1   Introduction

In recent years, systems with unconventional symmetries have attracted a lot of attention due to the wealth of exotic phenomena they display. In particular, the role of multipole-moment and subsystem symmetries have been explored extensively. Both of these have been shown to lead to exotic low-energy features including fracton phases of matter, fractal quantum spin liquids and quantum criticality, Bose surfaces and UV/IR-mixing [1–12]; as well as to unusual non-equilibrium behavior, such as unconventional transport [13–19] and Hilbert space fragmentation [20–22, 22–25, 25–29]. Dipole-conservation naturally appears as an approximate symmetry of systems subject to a strong tilted potential, and some of the aforementioned phenomena have been observed experimentally in this context [30–34].

Fragmentation in particular, appears naturally in systems that conserve multipole moments [20, 21]. It manifests as an exponentially large number of dynamically disconnected sectors even after resolving the global conserved quantities. Such fragmentation (or, in the parlance of classical stochastic dynamics, reducibility [35]) comes in multiple flavors. One can, for example, distinguish *weak* and *strong* fragmentation, which give rise to distinct dynamical signatures [20, 25]. A particularly striking example of the latter was found in Refs. [20, 21, 36], where fragmentation was shown to result in localized dynamics that retains local memory of initial conditions. While such strong reducibility has long been known for kinetically constrained glassy models [35], the recently studied examples, originating from dipole-conserving systems, were restricted to one spatial dimension [25, 36].

In this paper we introduce a set of models, which we name "discrete Laplacian models". Their defining feature is a correlated hopping term that distributes particles symmetrically among all neighboring sites, resembling a discrete second derivative and generalizing the 1D model of Refs. [20, 21] to arbitrary lattices. We prove that the discrete Laplacian models all exhibit the same kind of localized behavior due to the strong fragmentation of their configuration spaces. This fragmentation originates from a combination of the correlated hopping of particles and a local constraint on the number of particles per site. Together these lead to a finite density of sites whose configuration remains frozen at all times, implying strong fragmentation.

Even away from the strongly fragmented limit, we find that these models exhibit localized behavior close to the boundaries of the system. We explain this by the presence of *spatially*

Figure 1: **Discrete Laplacian models.** Family of models where the elementary process corresponds to a simultaneous hopping of particles from a site to all its neighbors. Illustrated here for different two-dimensional and three dimensional lattices.

*modulated symmetries*, a concept introduced in Ref. [37]. We show the discrete Laplacian models posses conserved quantities that, related to solutions of the discrete Laplace equation, are localized near the boundaries. These include exponentially localized charges at the corners of the lattice, but also other quantities that are *algebraically* localized close to the boundary. In fact, we construct multiple families of such quantities, which decay with increasing powers of the distance away from the boundary. Just as in Ref. [37], we utilize Mazur's bound to show that these symmetries prevent the decay of correlations at the boundary, even when the bulk is not localized.

The idea of conserved quantities localized near the boundaries of a system bears resemblance to that of *strong zero modes* (SZM) in quantum spin chains [38]. We will show that a modification of the discrete Laplacian models in the quantum setting can lead to phenomena analogous to SZM in two dimensions, with a system that exhibits degeneracies throughout its spectrum with open, but not with closed boundaries.[1] Nevertheless, our boundary modes differ in important ways from those of Ref. [38], as we discuss in detail below.

The remainder of the paper is organized as follows. In Sec. 2 we introduce a version of a discrete Laplacian model on a 2D square lattice and provide numerical results for the behavior of its bulk and boundary correlations, which will motivate our subsequent investigations. In Sec. 3, we first generalize the model to arbitrary lattices and graphs and then provide a proof of localization in the case when the number of local configurations is strongly restricted. In Sec. 4 we turn to the construction of conserved charges localized at the boundaries and discuss their implications for boundary correlations. Finally, in Sec. 5 we consider quantum versions of the discrete Laplacian models and we show how they can be modified to exhibit a phenomenology similar to strong zero modes. We conclude in Sec. 6.

## 2 Motivating example: Correlations in a 2D discrete Laplacian model

We begin by discussing numerical results for a particular 2D model. In subsequent sections, we will analytically explain the observed behavior, while also generalizing the model to arbitrary lattices. We consider models of classical stochastic dynamics, which allow for large-scale numerical simulations [16, 17, 37]. The reason is that we mostly explore "classical" phenomena that relies entirely on symmetries that are common to both the classical and quantum cases. We discuss features specific to quantum Hamiltonians at the end of the paper, in Sec. 5.

---

[1]A different construction of SZM in two-dimensional systems appeared recently in Ref. [39]. SZM in the context of stochastic classical dynamics have been recently discussed in Ref. [40].

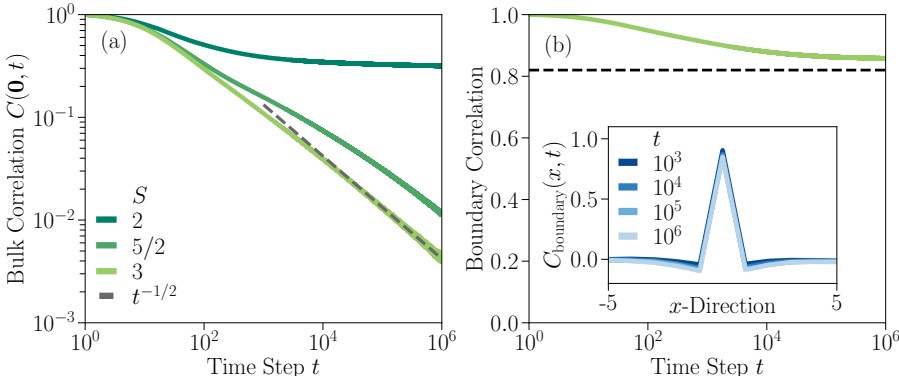

Figure 2: **Spin-spin correlations of $G_+$** (a) Spin-spin autocorrelation $C(\mathbf{0}, t)$ for linear system size $L = 300$ and different spins $S$. Data is averaged over $N = 1000$ random initial configurations and circuit realizations. When $S = 2$, the autocorrelation saturates to a constant value independent of system size, hence compatible with a strong fragmentation of the configuration space. For any spin $S > 2$ the autocorrelation decays as $t^{-1/2}$ (shown by the gray dashed line. (b) Boundary autocorrelation for $S = 3$. The black dashed line shows Mazur's bound as computed in Eq. (14). Inset: Spatial distribution along the $x$-axis defined as $C_{\text{boundary}}(x, t) \equiv C_{\mathbf{0}}(\mathbf{r} = (x, y = 0), t)$ for $L = 250$ and different times $t$. Unlike the weakly fragmented case, the correlation remains localized even at long times.

We consider a square lattice, where each site $\mathbf{r}$ hosts a classical spin $s_{\mathbf{r}}$ taking values in $s_{\mathbf{r}} \in \{-S, -S+1 \ldots, +S\}$ with (half-)integer $S$. We will equivalently refer to $s_{\mathbf{r}}$ as the amount of "charge" on site $\mathbf{r}$. The dynamics is generated by a set of local updates, or "gates", $G_+^{(\mathbf{r})}$, acting on site $\mathbf{r}$ and its four neighbors. The effect of a gate is to change $s_{\mathbf{r}} \to s_{\mathbf{r}} - 4$ and $s_{\mathbf{r}+\delta} \to s_{\mathbf{r}+\delta} + 1$ where $\boldsymbol{\delta} = \pm\mathbf{e}_x, \pm\mathbf{e}_y$ is any of the four elementary lattice vectors. In other words, it transfers one unit of charge from the central site $\mathbf{r}$ to all of its neighbors simultaneously (see Fig. 1a). This is well-defined for $S \geq 2$, and it is applied only if it does not lead to a violation of the local constraint $|s_{\mathbf{r}}| \leq S$. For future reference we summarize the effect of the gate $G_+$ by a set of integers as:

$$G_+ = \{n_{-1,0}, n_{0,1}, n_{0,0}, n_{0,-1}, n_{1,0}\} = \{1, 1, -4, 1, 1\}, \tag{1}$$

such that $s_{i,j} \to s_{i,j} + n_{i,j}$ under the effect of a gate applied at $\mathbf{r} = 0$. We also consider the inverse gate, which is obtained by replacing each $n_{i,j}$ with $-n_{i,j}$.

To turn these local updates into a stochastic evolution, in each time step we randomly tile the whole system with non-overlapping gates and apply the resulting layer (for additional details, see Ref. [37]). At each location, we choose randomly between the following three possibilities with equal probabilities: (i) apply the gate $G_+$, (ii) apply its inverse, or (iii) no update is made. The defined model conserves several multipole moments of the charge, namely: the total charge, both components of the dipole moment, and the traceless part of the quadrupole moment. With open boundary condition, it also has a large number of additional conserved quantities, as we shall discuss below in Sec. 4.

We study the dynamics of the system by investigating the behavior of "infinite temperature" connected charge-charge correlations, which are defined as

$$C_{\mathbf{r}_0}(\mathbf{r}, t) \equiv \frac{1}{D} \sum_{\mathbf{s}(0)} s_{\mathbf{r}_0}(0) \langle s_{\mathbf{r}}(t) \rangle_{\mathbf{s}(0)} - \left( \frac{1}{D} \sum_{\mathbf{s}(0)} s_{\mathbf{r}_0}(0) \right) \left( \frac{1}{D} \sum_{\mathbf{s}(0)} \langle s_{\mathbf{r}}(t) \rangle_{\mathbf{s}(0)} \right). \tag{2}$$

Here, we are uniformly sampling over all possible global initial configurations $\mathbf{s}(0)$ of the system, whose total number is given by $D = (2S+1)^N$ for a system of $N$ sites. $\mathbf{s}(t)$ is the time-

evolved configuration corresponding to a particular trajectory of the system, and the brackets denote averaging over different circuit realizations. We use $C(\mathbf{r}, t) \equiv C_{\mathbf{r}}(\mathbf{r}, t)$ to denote auto-correlations on the same site.

The expected generic late-time behavior for a system with the above multipole symmetries is a subdiffusive decay of correlations of the form $C(\mathbf{r}, t) \sim t^{-1/2}$ [14, 18, 41]. This behavior is indeed observed in our model when $S > 2$ as Fig. 2a shows. However, Fig. 2a also shows that for $S = 2$, bulk correlations saturate to a finite value that remains finite even in the thermo-dynamic limit (see App. A for a careful finite-size scaling analysis). We have also numerically confirmed that spatial correlations remain localized to a finite region. This is reminiscent of the behavior observed for $S = 1$ in the one-dimensional version of the same model [20], where it appeared as a result of Hilbert (or configuration in the classical model) space fragmentation. In Sec. 3 we argue that a similar explanation is valid here as well and it generalizes to a larger class of models, defined on arbitrary lattices and graphs.

While bulk correlations decay to zero for any $S > 2$ at infinite times, we find that correlations evaluated near the boundary of a system with open boundary conditions remain finite for *any S*, as shown in Fig. 2b. In Sec. 4, we will explain this by the presence of a set of additional conserved quantities localized at the edges of the system, similar to the behavior observed for certain 1D systems in Ref. [37]. We construct a hierarchy of conserved quantities that are power-law localized towards the bulk and utilize Mazur's bound to show that these lead to the finite boundary correlations observed above. This is consistent with the fact that we do not only observe finite boundary auto-correlations, but that in fact, the spatial correlation along the boundary defined as $C_{\text{boundary}}(x, t) \equiv C_0(\mathbf{r} = (x, y = 0), t)$ is also localized (see inset in Fig. 2b).

## 3 Localization from fragmentation

In this section, we first define a class of models, which we name "discrete Laplacian" models, that generalize both the 1D model studies in Ref. [20] and the 2D model discussed above in Sec. 2. We then go on to prove that all such models exhibit localized dynamics when the number of local configurations is sufficiently restricted.

### 3.1 General discrete Laplacian models

The model (1) introduced in the previous section can be thought of as a natural generalization of the 1D model studied in Refs. [20, 36]. In fact, as we now show, we can generalize this model to an arbitrary lattice in any spatial dimension, or even for an arbitrary graph. Consider a graph defined by vertices $V$ and edges $E$. Let $z_v$ denote the degree (number of neighbors) of vertex $v$ and we assume that the degrees are bounded: $z_v \leq z_0$, $\forall v \in V$. We assign a spin variable $s_v = -S_v, -S_v + 1, \ldots, +S_v$ to each site, where we now allow the local spin to vary with the vertex $v$. In the proofs of Sec. 3.2, we will find it easier to work with a shifted variable, $m_v = s_v + S_v$, which therefore takes values $0, 1, \ldots, M_v \equiv 2S_v$. We will refer to this latter variable as the "particle number" on vertex $v$.

Let us generalize the model in Eq. (1) and define the following local gate acting on a vertex $v$ and its neighbors $\mathcal{N}_v$, specified by the integers

$$n_{v'} = \begin{cases} -z_v, & \text{if } v' = v, \\ +1, & \text{if } v' \in \mathcal{N}_v. \end{cases} \tag{3}$$

The effect of this gate is to remove $z_v$ particles from $v$ and distribute them equally (one each) between its neighbors, leaving $v$ completely empty when $M_v = z_v$. This is illustrated in Fig. 1

for a number of different graphs corresponding to 2D and 3D lattices. We also define the inverse gate, which takes one particle from each of the neighbors of $v$ and collects them at $v$. As before, we only allow these gates to act if it does not lead to a violation of the constraint $0 \leq m_v \leq M_v$ for any vertex $v$. We refer to the set of models built from these elementary updates as *discrete Laplacian models*.[2]

While there are various ways in which these local updates rules can be turned into a stochastic evolution, the properties we discuss are largely independent of these details and follow directly from the constraints imposed by the structure of the gates (although we do assume detailed balance). We note that the discrete Laplacian updates are fairly natural to consider if one wants to impose a conservation of higher (in particular, dipole) moments while keeping interactions as local as possible.

### 3.2 Proof of localization

We now turn our attention to the case when the local constraints $M_v$ take on their minimal value, i.e., we set $M_v = z_v$ throughout this section. For the 2D model in Fig. 1a, we observed that in this case autocorrelations exhibit a finite saturation value, a signature of localization. We now prove that this is a generic feature of the discrete Laplacian models with $M_v = z_v$.

As discussed above, there are two elementary processes that can occur: a site can 'fire', distributing a particle to each of its neighbors (i.e., $G(v) = -z_v$) or it can 'anti-fire', receiving a particle from all neighbors simultaneously ($G(v) = +z_v$). As we emphasized, these updates are allowed only if they do not lead to a violation of the constraints $m_v \in [0, z_v]$. To specify the dynamics, we consider stochastic evolutions where these two types of local updates are applied according to some random rules, at integer times, defining a Markov process with transition matrix $T$. We assume that this Markov process is *reversible*, $T_{\mathbf{m},\mathbf{m}'} = T_{\mathbf{m}',\mathbf{m}}$. The space of particle configurations splits into various connected components; due to the reversibility of the dynamics, for an initial condition belonging to connected component $\mathcal{C}$, there is a unique stationary distribution, which is uniform over all $\mathbf{m} \in \mathcal{C}$ [43].

Our strategy to prove localization is as follows. We fix a vertex $v$ and we identify connected components such that $m_v$ takes the same value for all $\mathbf{m} \in \mathcal{C}$; in particular the values $m_v = 0$ or $m_v = z_v$ (its minimum and maximum). Any initial configuration from such a $\mathcal{C}$ will lead to a finite contribution to the connected autocorrelation $C_v(t)$. What we then need to prove is that a finite fraction of all initial configurations belong to one of these connected components.

In particular, let $D(\mathcal{C}) = |\mathcal{C}|$ denote the number of configurations belonging to $\mathcal{C}$ and $D_{m_v}(\mathcal{C})$ denote the number of those where the vertex $v$ is in the state $m_v$ (obviously, $\sum_{m_v} D_{m_v}(\mathcal{C}) = D(\mathcal{C})$). We can define the average value of $m_v$ within the sector $\mathcal{C}$ as $\overline{m}_v(\mathcal{C}) \equiv \sum_{m_v} m_v \frac{D_{m_v}(\mathcal{C})}{D(\mathcal{C})}$.

Let us write the average over random trajectories as $\langle m_v(t) \rangle_{\mathbf{m}(0)} = \sum_{\mathbf{m}} p_{\mathbf{m}(0)}(\mathbf{m}, t) m_v$, where $p_{\mathbf{m}(0)}(\mathbf{m}, t) = \left( T^t \right)_{\mathbf{m},\mathbf{m}_0}$ is the probability distribution over possible spin configurations, conditioned on a given initial configuration $\mathbf{m}(0)$. Let us now focus on $\langle m_v(t) \rangle_{\mathbf{m}(0)}$ for a particular initial configuration belonging to a specific connected component $\mathcal{C}$. Restricted to this, the dynamics is by definition irreducible. Since we also assumed reversibility, this implies a unique stationary distribution, which is uniform over all $\mathbf{m} \in \mathcal{C}$ [43]. We therefore have

$$\lim_{\tau \to \infty} \frac{1}{\tau} \sum_{t=0}^{\tau} p_{\mathbf{m}(0)}(\mathbf{m}, t) = \frac{1}{D(\mathcal{C})} \delta_{\mathcal{C}}(\mathbf{m}) \equiv p_{\mathcal{C}}(\mathbf{m}), \tag{4}$$

---

[2]We note that these are closely related to the problem of *chip-firing* studied in the mathematical literature [42]. However, our models are different in that they allow for the aforementioned inverse gates and in that there is a local constraint $s_v \leq M_v$.

with $\delta_{\mathcal{C}}(\mathbf{m}) = 1$ if $\mathbf{m} \in \mathcal{C}$ and zero otherwise. Using this, the infinite time-averaged expectation value is $\overline{\langle m_v(t) \rangle_{\mathbf{m}(0)}} = \overline{m}_v(\mathcal{C})$, which is independent of $\mathbf{m}(0)$ within the same $\mathcal{C}$. Plugging this back into Eq. (2) we get

$$\overline{C_v} \equiv \lim_{\tau \to \infty} \frac{1}{\tau} \sum_{t=0}^{\tau} C_v(\tau) = \sum_{\mathcal{C}} \frac{D(\mathcal{C})}{D} (\overline{m}_v(\mathcal{C}) - \overline{m}_v)^2 \,, \tag{5}$$

where $D = \sum_{\mathcal{C}} D(\mathcal{C}) = \prod_v (z_v + 1)$ is the total number of configurations and $\overline{m}_v = z_v/2$ is the overall ("infinite temperature") average particle number.[3] The key point about this formula is that every term in the sum on the right hand side is non-negative. Therefore calculating the contribution from any subset of the terms in the sum of Eq. (5) provides a strict lower bound for the time-averaged autocorrelations.

We will construct the appropriate set of connected components by identifying local spin configurations where certain spins remain frozen in the original state at all times, independently of the configuration in the rest of the system. The same approach can be used to demonstrate strong fragmentation in kinetically constrained spin systems, such as certain variants of the Fredrickson-Andersen model [35]. However, in our case, proving the existence of such frozen blocks is considerably more involved, and it will take up the rest of this section. We begin by proving a useful lemma:

**Lemma 1.** *Assume a vertex $v$ fires twice at two different (discrete) times $t'-1$ and $t+1$ ($t > t'$) such that it does not anti-fire (on net) between them. Then all neighbors of $v$ must have fired at least once sometime in the time window $[t', t]$.*

*Proof.* We want to keep track of how many times each site fires/anti-fires as the dynamics progresses. Let us define the *net number* of firings at a vertex $F_v(t, t')$ as the number of times $v$ fires in a time interval $[t', t]$ *minus* the number of times it anti-fires in the same time interval.

In the situation above we have $F_v(t+1, t'-1) = +2$ and $F_v(\tau, \tau') = 0$ for any $t' \le \tau' < \tau \le t$. In particular, $F_v(t, t') = 0$. Now we assume that $v$ has some neighbor $v'$ that does *not* fire in this time interval (more precisely, we only need the weaker condition that it does not fire *on net*, i.e., $F_{v'}(t, t') \le 0$.) and derive a contradiction.

Note that between the two firings, $v$ has to 're-charge', $\Delta m_v(t, t') \equiv m_v(t) - m_v(t') = z_v$. On the other hand, we can rewrite this as the total charge flowing into site $v$:

$$\Delta m_v(t, t') = \sum_{v' \in \mathcal{N}_v} F_{v'}(t, t') - z_v F_v(t, t') = \sum_{v' \in \mathcal{N}_v} F_{v'}(t, t') = z_v \,. \tag{6}$$

That is, the charge needed for the second firing has to come from the firing of neighboring sites. If we assume that one of the neighbors does not fire, then another one has to fire at least twice: $F_{v_1}(t, t') \ge 2$ for some $v_1 \in \mathcal{N}_v$.

Now we can run this argument recursively. Let $t_1$ be the time when $v_1$ fires for the second time and $t_1'$ the last time it fired before that. Between these two times, $v_1$ needs to re-charge, which it can only do if its neighbors fire. However, since $t' < t_1' < t_1 < t$ we have from our previous assumption that $F_v(t_1, t_1') = 0$. Therefore $v_1$ needs to have some *different* neighbor, $v_2$ that fires at least twice at some times $t_2'$ and $t_2$ between $t_1'$ and $t_1$ and so forth. We thus end up with an infinite regress: before our original site could fire for the second time, one of its neighbors has to fire twice, but then one of *its* neighbors has to fire twice etc, making this process impossible (for a sketch of this on the square lattice, see Fig. 3a). The only resolution is to allow *all* neighbors of $v$ to fire at least once between $t'$ and $t$.

$\square$

---

[3]Note that the bound itself is also insensitive to the shift of variables from $s_v$ to $m_v$.

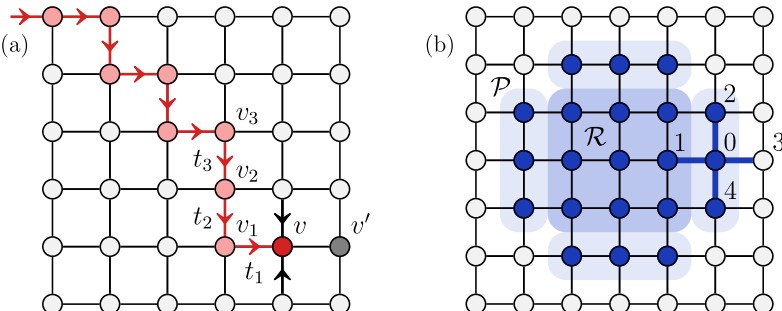

Figure 3: **Proof of localization.** (a) Sketch of the argument leading to lemma 1. We want site 0 (dark red) to fire twice, while not allowing its neighbor to the right (gray) to fire in-between. This requires another neighbor (site 1) to fire twice, leading to an infinite regress shown by the red arrows. (b) Construction of a cluster of frozen sites. Blue sites are taken to have zero particles $m_v = 0$. The light blue sites on the outside layer can change but the inner ones remain frozen forever, independently of the initial configuration on all other sites.

We can then use this property to construct frozen blocks of sites that lead to fragmentation and localized dynamics. In particular, we have

**Corollary 1.** *Take some set of sites $\mathcal{R}$ with the property that for any $v \in \mathcal{R}$ there is some neighbor $v' \in \mathcal{N}_v$ that is also in $\mathcal{R}$. Let $\overline{\mathcal{R}}$ denote the set of vertices in $\mathcal{R}$ along with all their neighbors, $\overline{\mathcal{R}} \equiv \mathcal{R} \cup \left( \bigcup_{v \in \mathcal{R}} \mathcal{N}_v \right)$. Consider a configuration where all sites in $\overline{\mathcal{R}}$ have $m_v = 0$. Then the sites in $\mathcal{R}$ will remain frozen forever, independently of what the initial configuration is outside of $\overline{\mathcal{R}}$.*

*Proof.* An example of this situation for a square lattice is shown in Fig. 3b for the 2D square lattice, with dark blue sites belonging to $\mathcal{R}$ and light blue ones to the "padding region" $\mathcal{P} \equiv \overline{\mathcal{R}} \backslash \mathcal{R}$. We again use a proof by contradiction. Let us assume that at least one of the sites in $\mathcal{R}$ can change. Let us denote by $v$ the first one to do so (site labeled '1' in Fig. 3b). The only way for $v$ to change is if one of its neighbors in $\mathcal{P}$ fires. Let us denote this site by $v' \in \mathcal{N}_v \cap \mathcal{P}$ (site labeled '0' in the figure).

Since $v'$ starts in its lowest state, it must have been completely 'charged up' (i.e., $m_{v'} = z_{v'}$) beforehand. Site $v'$ could not have anti-fired up to this point, since it has a neighbor (site $v$) which has no particles. Therefore the charge must have come from the firing of its other neighbors. Since $v$ is frozen, only the remaining $z_{v'} - 1$ neighbors (sites $2, 3, 4$ in the figure) have contributed to charge $v$ up. Hence, one of them needed to fire at least twice. However, due to the previous theorem, this is in contradiction with the fact that $v'$ did not yet fired, which finishes the proof. $\square$

We can now combine this corollary with Eq. (5) to arrive at

**Theorem 1.** *The time-averaged "infinite temperature" auto-correlations are finite in the thermodynamic limit. In particular $\overline{C_v} \geq (z_v/2)^2 (z_0 + 1)^{-2z_0}$.*

*Proof.* Let us take $\mathcal{R}$ to be a set of two sites, including $v$ and one of its neighbors and consider the set of configurations such that all sites $v' \in \overline{\mathcal{R}}$ have $m_{v'} = 0$. The number of such configurations is $D / \left( \prod_{v' \in \overline{\mathcal{R}}} (z_{v'} + 1) \right) \geq D / (z_0 + 1)^{2z_0}$, where we used that $\overline{\mathcal{R}}$ contains at most $2z_0$ sites. Let us now also include all other configurations that are connected to one of these by the dynamics, forming a set of connected components $\mathcal{C}$. For all such $\mathcal{C}$, it follows from the corollary that $\overline{m}_v(\mathcal{C}) = 0$, since $v$ is frozen. Therefore the total contribution from these to

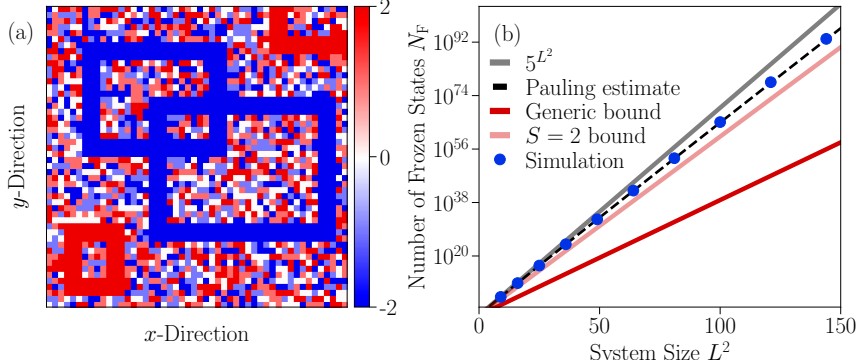

Figure 4: **Fragmentation.** (a): Construction of various sectors in configuration space using corollary 1, with inert walls splitting the lattice into disconnected regions. (b) Estimates for the number of completely frozen configurations, along with a Pauling estimate (see main text). Numerical simulations use system sizes with $L \leq 12$.

the RHS of Eq. (5) is $(z_v/2)^2 \left( \sum'_{\mathcal{C}} D(\mathcal{C}) \right)/D \geq (z_v/2)^2/(z_0+1)^{2z_0}$. We hence conclude that the time-averaged autocorrelations must remain finite. In particular, for the square lattice model in Eq. 1 which has $z_v = 4 \, \forall v$, we find $\overline{C_v} \geq 4/5^8$. $\qquad \square$

Theorem 1 is the main result of this section. It shows that the dynamics is localized, in the sense that a finite amount of local memory of initial conditions is preserved forever.

### 3.3 Strong fragmentation of the configuration space

Let us now discuss what our results imply about the fragmented structure of the space of particle number configurations, i.e. the structure of the connected components $\mathcal{C}$. For simplicity, we focus on a case where the system is defined on a $d$-dimensional lattice, so there is a notion of a linear size $L$, with the number of sites growing as $\sim L^d$. We also take periodic boundary conditions, so that there are no additional global symmetries beyond the multipole moments mentioned above.

Ref. [20] distinguished two types of fragmentation, labeled *weak* and *strong*. For the former, once the values of global conserved quantities (in our case, charge and its various multipole moments) are fixed, there is a dominant $\mathcal{C}$ that contains almost all configurations in the thermodynamic limit. In particular, this means that the size of the largest connected component has a size $D/\text{poly}(L)$. For strong fragmentation, on the other hand, the dimension of the largest $\mathcal{C}$ is an exponentially small fraction of $D$. The one-dimensional version of the discrete Laplacian model was found to exhibit strong fragmentation for $S = 1$ [20], and the localization of autocorrelations was derived explicitly from a complete classification of the fragmented components in Ref. [36].

Here, we proved localization more directly, without having to construct the full set of connected components. Nevertheless, our results do imply strong fragmentation. What we have proven above is that the system exhibits a finite *frozen site density* [44]: any given site has a non-vanishing probability of having its local state preserved by the dynamics at all times. It follows from this that the probability that a randomly chosen configuration contains a finite fraction of such frozen sites, approaches 1 in the limit of large system sizes. The presence of a finite fraction of frozen sites means that the number of configurations connected to this initial state is exponentially small compared to $D$. This implies strong fragmentation. More previsely, as we prove in the following

**Theorem 2.** *Consider a generalized discrete Laplacian model defined on a d-dimensional hypercubic lattice of linear size L, such that each site has $M = z + 1$ possible states, with $z$ the coordination number of the lattice. Then the space of particle number configurations is strongly fragmented in the sense that the size of the largest connected component divided by $D = M^{L^d}$ goes to zero exponentially with $L^d$.*

*Proof.* If a configuration has $n$ frozen sites, i.e. sites whose state cannot evolve, then it is connected at most to $M^{L^d-n}$ other configurations. This is exponentially small compared to $D$ if $n/L^d$ is finite in the limit of large $L$. Therefore, any putative large component that is *not* exponentially small must contain only a vanishing fraction of frozen sites. If we can show that the total number of configurations where $n$ does not grow proportionally to $L^d$ is itself exponentially small compared to $D$ then we have ruled out the possibility of a large (not exponentially small) connected component.

Let us partition the lattice into regions ("boxes") of linear size $\ell$. The number of such boxes is $N_\ell = (L/\ell)^d$. This means that each box has $D_\ell = M^{\ell^d}$ configurations, while the whole system has $D = M^{L^d} = D_\ell^{N_\ell}$. Corollary 1 shows that we can construct finite-sized patterns of maximally polarized spins where some sites (those on the "inside", i.e., the region $\mathcal{R}$ in Fig. 3) are frozen independently of what the configuration is outside of the pattern. If we choose $\ell$ to be large enough (i.e., $\ell \geq 4$ for a hypercubic lattice), then each box can contain such a frozen pattern. This will happen with a finite probability $p_\ell$ that depends on $\ell$ but not on the overall system size $L$. In particular, let $F_\ell$ be the number of configurations of a box that contain a frozen pattern, then $p_\ell = F_\ell/D_\ell$. Let us denote by $G_\ell = D_\ell - F_\ell$ the number of remaining configurations and $q_\ell = G_\ell/D_\ell < 1$ their probability.

We now ask the question: what is the probability that a randomly chosen initial spin configuration contains at most $n$ frozen sites? Let us denote this quantity $P_{\leq n}$. We want to show that $P_{\leq n}$ will be exponentially small in the large $L$ limit, unless $n$ is a finite fraction of $L^d$ (in particular, it should be at least $p_\ell L^d$). To do so, we note that we can upper bound it in the following way. Let us choose $m$ boxes which contain a frozen pattern, while the remaining $N_\ell - m$ boxes have no frozen patterns. Clearly, any such state has at least $m$ frozen sites, so the only contributions to $P_{\leq n}$ come from $m \leq n$. This gives the following upper bound:

$$P_{\leq n} \leq \sum_{m=0}^{n} \binom{N_\ell}{m} \frac{F_\ell^m G_\ell^{N_\ell-m}}{D_\ell^{N_\ell}} = \sum_{m=0}^{n} \binom{N_\ell}{m} p_\ell^m q_\ell^{N_\ell-m}. \tag{7}$$

The expression on the right hand side is the cumulative distribution function of the binomial distribution. At large $N_\ell$, the binomial distribution is increasingly sharply peaked around its mean value, $\bar{n} = p_\ell N_\ell$, and the cumulative probability at values of $n$ much smaller than $\bar{n}$ (i.e., outside a window of size $\propto N_\ell^{1/2}$) will be exponentially suppressed in $N_\ell$. In particular, the probability of configurations where the number of frozen sites is not a finite fraction of $L^d$ goes to zero exponentially in the thermodynamic limit. $\square$

In fact, there are many more connected components beyond those that we used in the proof of theorem 1, some of which we can also construct by using Corollary 1. For example, we can take the region $\mathcal{R}$ to be a closed surface, meaning that it separates the graph into two regions – an "inside" and an "outside" – such that any path connecting them must go through $\mathcal{R}$. Then the two sides of $\mathcal{R}$ will remain disconnected by the dynamics (an example of this behavior is shown in Fig. 4a). The region $\mathcal{R}$ in this case plays the same role as the "inert regions" discussed for the 1D model in Refs. [20, 21]. One could also put any frozen configuration in the inside of $\mathcal{R}$ to get some large frozen island, which would give additional contributions to the autocorrelations as well.[4]

---

[4]Note also that we could re-run corollary 1 for a case where we fix the sites in $\mathcal{R}$ to have maximal values, $m_v = z_v$,

Another quantity related to fragmentation is the total number of *frozen states*, i.e., spin configurations where every site is frozen. Numerically, we can estimate their number by randomly sampling configurations and checking if they are frozen: the results for the 2D model (1) with $S = 2$ are shown by the dots in Fig. 4b. We can also give an analytical estimate using Pauling's method, which yields that the number of frozen states, $N_F$, scales as $\ln(N_F) \approx \ln(D) + (L-2)^2 \ln\left(1 - \frac{2^9}{5^5}\right)$ (see additional details in App. B). While this estimate is non-rigorous, it fits the numerical results very well (see black dashed line in Fig. 4b). For the particular choice $S = 2$, we can bound $N_F$ by noting that only states with highest or lowest possible spins can evolve in time, finding the lower bound $3^{L^2}$ (pale red line in Fig. 4b). Nevertheless, we can also derive a less tight but rigorous lower bound which holds for all integer and half-integer $S$. Let us consider a tiling of the 2D lattice with some unit cell and fix the configuration of spins on some (proper) subset of sites in each unit cell in such a way that these prevent all sites from firing (or anti-firing), no matter what configuration we put on the remaining sites. If the fraction of fixed sites is $\phi$, it directly follows that $N_F > (2S+1)^{(1-\phi)L^2}$. There are several possible tilings that yield lattices that remain frozen at all times. Fig. 11 in App. B shows one particular example, using a $3 \times 3$ unit cell with four fixed sites. The finite fraction is therefore $\phi = 4/9$ and the lower bound becomes $N_F > (2S+1)^{\frac{5}{9}L^2}$. This is sufficient to prove that the number of frozen states scales exponentially with $L^2$ (see red line in Fig. 4b for $S = 2$) for any $S$, even away from the limit of $S = 2$ where our proof of strong fragmentation applies.

## 4 Spatially modulated charges and boundary correlations

As observed in Fig. 2a, autocorrelations in the 2D model defined in Eq. (1) decay to zero in the bulk for any spin $S > 2$. However, as panel (b) of the same figure shows, this is not true for correlations near the boundary, when the system is defined with open boundary conditions. Here, we explain this fact in terms of additional conserved quantities that the model possesses in this case.

### 4.1 Recursion relation: Discrete Laplace equation

In general, we can look for spatially modulated conserved quantities of the form $\mathcal{Q}_{\{\alpha_\mathbf{r}\}} = \sum_\mathbf{r} \alpha_\mathbf{r} s_\mathbf{r}$. For the model in Eq. (1), this ansatz leads to the following recurrence relation, whose solutions correspond to conserved quantities:

$$4\alpha_{i,j} - \alpha_{i+1,j} - \alpha_{i-1,j} - \alpha_{i,j+1} - \alpha_{i,j-1} = 0. \tag{8}$$

With open boundaries, one can always solve the recurrence equation by specifying some set of boundary values $\alpha_{i,j}^B$ and then propagating them to the bulk. Such solutions give rise to a large number of additional conservation laws. As we show below, these are localized near the boundary and lead to the aforementioned long-lived correlations there, while their effect on bulk dynamics is negligible in the thermodynamic limit.

To prove that this is the case, we need to solve Eq. (8) for specified boundary values of $\alpha$'s for $(i,j) \in B$, i.e., $\alpha_{0,j}, \alpha_{L+1,j}, \alpha_{i,0}, \alpha_{i,L+1}$, where we distinguish between interior sites $i, j \in \{1, \ldots, L\}$, that belong to the bulk of the system $D$, and those at the boundary $B$. To make the solution of the recurrence relation more apparent, we rewrite the equation in a slightly

---

instead of $m_v = 0$. This produces a different set of connected components and corresponding contributions to the autocorrelation.

different form:

$$\begin{cases} \alpha_{i,j} = \frac{1}{4} \left( \alpha_{i+1,j} + \alpha_{i-1,j} + \alpha_{i,j+1} + \alpha_{i,j-1} \right), & \text{for } (i,j) \in D, \\ \alpha_{i,j} = \alpha_{i,j}^{\text{B}}, & \text{for } (i,j) \in B. \end{cases} \tag{9}$$

Hence we see that the value at site $(i,j)$ in the bulk, is given by the average value of the four neighboring sites. This equation is a discrete version of the Laplace equation $((\partial_x^2 + \partial_y^2)\alpha(x,y) = 0)$ with Dirichlet boundary conditions for a square tiling. Its solutions are known as *discrete harmonics* (see e.g., Ref. [45]). There are two important properties of (discrete) harmonic functions that we will use:

1. A (discrete) harmonic function defined on $\mathcal{L}$ takes its maximum $M$ and minimum $m$ values at the boundary $B$. E.g., if $\alpha_{i,j}^{\text{B}} \in \{0,1\}$ then $0 \leq \alpha_{i,j} \leq 1$ for all points in the bulk.

2. Using this fact, it follows that the solution is *unique*: given two harmonic functions $f, g$ on $\mathcal{L}$ such that $f = g$ on $B$ implies $f_{i,j} = g_{i,j}$ for all $(i,j) \in \mathcal{L}$.

One general way of solving a higher-dimensional linear recurrence equation, and in particular the Dirichlet's problem in Eq. (9), is using separation of variables $\alpha_{i,j} = X_i Y_j$. With this, Eq. (8) simplifies to solving the one-dimensional recurrences

$$\begin{cases} X_{i+1} - 2X_i + X_{i-1} = \lambda X_i, \\ Y_{j+1} - 2Y_j + Y_{j-1} = -\lambda Y_j, \end{cases} \approx \begin{cases} X''(x) = \lambda X(x), \\ Y''(y) = -\lambda Y(y), \end{cases} \tag{10}$$

where we also indicated the corresponding continuum analogues. $\lambda$ here is a constant whose possible values are restricted by the choice of boundary conditions. In particular, due to the linearity of the problem, we can decompose the Dirichlet's problem into four independent ones, with $\alpha$ vanishing along 3 out of the 4 boundaries in each case. This leads to a quantization condition on $\lambda$ and the general solution will be a linear combination of such fundamental solutions. For now, we keep $\lambda$ as an arbitrary parameter and consider what the form of the solutions admitted by Eqs. (10) is. See additional details in Appendix D and in Ref. [46].

We can solve Eqs. (10) by finding the roots of the associated characteristic polynomials $r^2 - (2 \pm \lambda)r + 1 = 0$ where the two signs correspond to the equations for $X$ and $Y$ respectively. For each equation, the two roots are inverses of each other and are either real $(\xi_{x,y})$ or pure complex phases $(e^{ik_{x,y}})$, depending on the value of $\lambda$. Overall, we can identify three main types of solutions: (i) When $\lambda = 0$, we have $\xi_x = \xi_y = 0$; this case contains the multipole conserved quantities discussed earlier.[5] (ii) When $0 < |\lambda| < 4$, one of the solutions is real while the other one is complex, e.g., $\alpha_{i,j} \propto (\xi_x)^i e^{ik_y j}$. These correspond to conserved quantities that are exponentially localized in one dimension while being fully delocalized in the other. (iii) When $|\lambda| > 4$, both solutions are real $\alpha_{i,j} = (\xi_x)^i (\xi_y)^j$, and hence can be exponentially localized near one of the corners of the system, depending on the modulus of $\xi_x$ and $\xi_y$.

Apart from giving insight into the family of possible conserved quantities, these fundamental solutions can be combined to obtain the unique solution corresponding to a choice of boundary values $\alpha_{i,j}^{\text{B}}$. Instead of writing that general expression, we notice that close-form solution of equation (9) is already known in the context of two-dimensional (unbiased) random walks [47]. This is given by

$$\alpha_{i,j} = \sum_{a=1}^{L} \alpha_{a,L+1}^{\text{B}} T_{a,L}(i,j) + \sum_{a=1}^{L} \alpha_{a,0}^{\text{B}} T_{a,1}(i,j) + \sum_{b=1}^{L} \alpha_{0,b}^{\text{B}} T_{1,b}(i,j) + \sum_{b=1}^{L} \alpha_{L+1,b}^{\text{B}} T_{L,b}(i,j), \tag{11}$$

---

[5]The one exception is the $(i^2 - j^2)$ quadratic moment, which does not factorize in the horizontal and vertical directions. We can recover it by superposing fundamental solutions or, alternatively, by writing the recurrence relation in terms of the center of mass and relative coordinates, as $\alpha_{i,j} = X_{i+j} Y_{i-j}$.

where each sum corresponds to one of the four boundaries with the respective boundary values $\alpha^{\mathrm{B}}_{i,j}$ and

$$T_{a,b}(i,j) = \frac{2}{(L+1)^2} \sum_{r=1}^{L} \sum_{s=1}^{L} \frac{\sin(\frac{ir\pi}{L+1})\sin(\frac{js\pi}{L+1})\sin(\frac{ar\pi}{L+1})\sin(\frac{bs\pi}{L+1})}{2 - \cos(\frac{r\pi}{L+1}) - \cos(\frac{s\pi}{L+1})}. \tag{12}$$

Hence, $\alpha_{i,j}$ is given by the (discrete) convolution of $T_{a,b}(i,j)$ with $\alpha^{\mathrm{B}}_{i,j}$. $T_{a,b}(i,j)$ is the discrete analogue of the Poisson kernel, whose convolution with the boundary condition solves the continuum Dirichlet problem [48]. We will use this fact to study the behavior of the solutions of the discrete recurrence relation Eq. (8) in the following section.

While Eqs. (11) and (12) provide a way to construct the exact solutions for any boundary condition, in practice we instead solve the recurrence relation numerically, applying an iterative method outlined in App. C. This has the advantage that it can be easily generalized to other cases where the exact kernel is not known, some of which are studied in Sec. 4.3.

## 4.2 Boundary localized charges and Mazur's bound

A powerful tool to analytically prove finite boundary correlations is given by Mazur's bound [49] $M_{i,j}$, which lower bounds the infinite time-average auto-correlations by the overlap with a set of conserved quantities $\mathcal{Q}_a$. More explicitly this is

$$\lim_{T \to \infty} \frac{1}{T} \int_0^T \mathrm{d}t \, \langle s_{i,j}(t)s_{i,j}(0) \rangle \geq M_{i,j}, \tag{13}$$

at any site $(i,j)$. To define the bound, we introduce an inner product between observables as $\langle A, B \rangle \equiv \langle AB \rangle$ where $\langle A \rangle = \frac{1}{|\mathcal{C}|} \sum_{\{s_{i,j}\} \in \mathcal{C}} A(s_{i,j})$ is the "infinite-temperature" average. With this definition, the bound can be written as

$$M_{i,j} \equiv \sum_{a,b} \langle s_{i,j}, \mathcal{Q}_a \rangle (K^{-1})_{a,b} \langle \mathcal{Q}_b, s_{i,j} \rangle. \tag{14}$$

Here, $K$ is a positive-definite matrix with elements $K_{a,b} = \langle \mathcal{Q}_a, \mathcal{Q}_b \rangle$. If one includes only a single conserved quantity $\mathcal{Q}_{\{\alpha_{\mathbf{r}}\}} = \sum_{\mathbf{r}} \alpha_{\mathbf{r}} s_{\mathbf{r}}$, then the expression simplifies to

$$M_{i,j} \equiv \frac{|\langle s_{i,j}, \mathcal{Q}_{\{\alpha_{\mathbf{r}}\}} \rangle|^2}{\langle \mathcal{Q}_{\{\alpha_{\mathbf{r}}\}}, \mathcal{Q}_{\{\alpha_{\mathbf{r}}\}} \rangle}, \tag{15}$$

where $\langle \mathcal{Q}_{\{\alpha_{\mathbf{r}}\}}, \mathcal{Q}_{\{\alpha_{\mathbf{r}}\}} \rangle$ is simply given by

$$\langle \mathcal{Q}_{\{\alpha_{\mathbf{r}}\}}, \mathcal{Q}_{\{\alpha_{\mathbf{r}}\}} \rangle = \frac{S(S+1)}{3} \sum_{i,j} (\alpha_{i,j})^2 = \frac{S(S+1)}{3} \|\alpha\|_2^2, \tag{16}$$

i.e., proportional to the 2-norm of $\alpha$. Hence, if one finds even a single conserved quantity that is strongly localized around some particular site at the boundary of the system and whose norm remains finite in the thermodynamic limit, this will give rise to finite boundary correlations. Alternatively, even if individual charges are not sufficiently well-localized, they can still be combined to give rise to a finite value on the RHS of Eq. (14).

### 4.2.1 Corner charges: Exponential localization

We first consider the situation near the corners of the square lattice. On a square lattice, the corner sites are completely decoupled under the dynamics, and hence provide a trivially conserved boundary charge. Let us ignore these, and consider instead the region close to the corner with coordinates $\mathbf{r}_0 = (0,0)$.

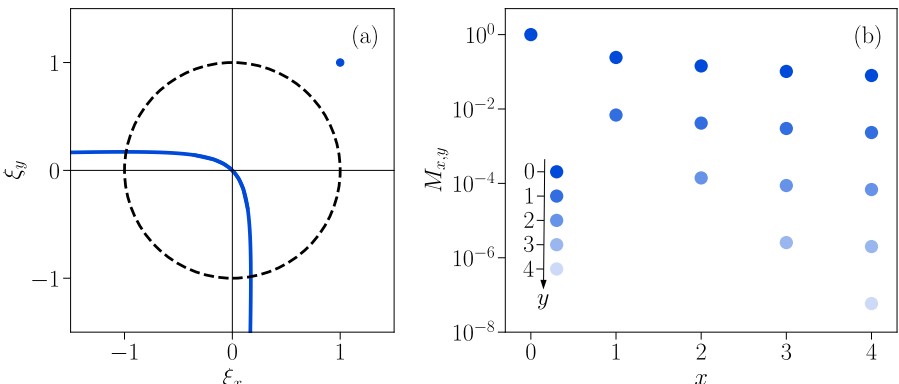

Figure 5: **Exponentially localized charges at the corners.** (a) Solutions of Eq. (17). The black dashed line shows the unit circle. (b) Largest value of Mazur's bound in Eq. (18) over the set of solutions plotted in panel (a), at different coordinates $(x, y)$.

As we noted in Sec. 4.1, the recursion relation has fundamental solutions of the form $\alpha_{ij} \propto (\xi_x)^i (\xi_y)^j$. This suggests conserved quantities that are exponentially localized near one of the four corners of the system, similar to the exponentially localized quantities found for certain 1D systems in Ref. [37]. Indeed, we can plug the above ansatz directly into the original recursion relation (8), which then turns into

$$\xi_x + \frac{1}{\xi_x} + \xi_y + \frac{1}{\xi_y} = 4. \tag{17}$$

Solutions to this, restricted to the domain $|\xi_x|, |\xi_y| < 1$ exist (see Fig. 5a) and provide exponentially localized modes at the corner $(0, 0)$.[6]

We can plug these conserved quantities into Mazur's bound (15) at a site $(x, y)$ close to the corner (i.e.,. $|x|, |y|$ not scaling with $L$) and including only this conserved quantity to find

$$M_{x,y} = \frac{S(S+1)}{3} \frac{(\alpha_{x,y})^2}{\sum_{i,j} (\alpha_{i,j})^2} \longrightarrow \frac{S(S+1)}{3} (\xi_x)^{2x} (\xi_y)^{2y} (1 - (\xi_x)^2)(1 - (\xi_y)^2), \tag{18}$$

in the limit $L \to \infty$. Therefore, as long as the site $(x, y)$ is at finite distance from the corner, Mazur's bound will be finite, which in turn shows that the time-average boundary correlation saturates to a finite value. The optimal lower bound can be found by maximizing the expression on the right hand side of Eq. (18) over the set of solutions of the equation (17). The optimal bounds thus obtained for different coordinates $(x, y)$ are shown in Fig. 5b.

### 4.2.2 Mid-boundary charges: Power-law localization

In Sec. 4.1 we found that there exist fundamental solutions of the recursion equation of the form $\alpha_{i,j} \propto (\xi_x)^i e^{ik_y j}$, i.e. exponentially localized in one direction and plane-wave-like in the other. These are symmetries localized at the boundary. However, they are not normalizable and do not immediately yield a finite Mazur bound. Here we instead ask whether we can find solutions that are localized in all directions. To answer this, we return to Eq. (9), and recall that it is a lattice discretization of the continuum Laplace equation. Motivated by this, we first consider the problem in the continuum, which will guide us in constructing new boundary charges and explaining their localization properties.

---

[6]Notice that if $(\xi_x, \xi_y)$ is a solution of Eq. (17), so are $(\frac{1}{\xi_x}, \frac{1}{\xi_y})$, $(\frac{1}{\xi_x}, \xi_y)$ and $(\xi_x, \frac{1}{\xi_y})$. These correspond to charges localized at the other three other corners of a finite lattice.

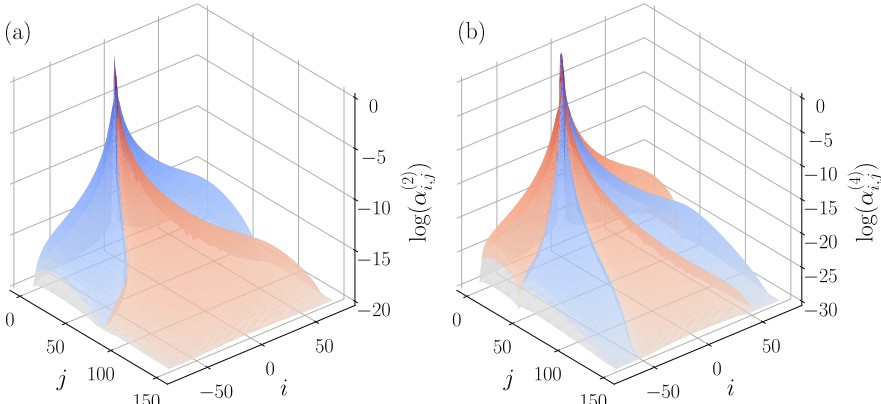

Figure 6: **Boundary charges with $n > 1$.** Solutions of Eq. (9) with $n = 2$ (panel (a)) and $n = 4$ (panel (b)), which decay towards the bulk as $\alpha^{(2)}_{0,j} \sim j^{-3}$ and $\alpha^{(4)}_{0,j} \sim j^{-5}$ respectively. Red and blue correspond to positive and negative values of $\alpha^{(2)}_{i,j}$.

As we are interested in the long-distance behavior of boundary charges, we consider Laplace's equation on a semi-infinite plane, i.e., on the domain $(x, y) \in (-\infty, \infty) \times (0, \infty)$.[7] That is,

$$\begin{cases} (\partial_x^2 + \partial_y^2)\alpha(x, y) = 0, \\ \alpha^{\mathrm{B}}(x, 0) = f(x), \quad \alpha(x \to \pm\infty, y) = 0, \quad \alpha(x, y \to +\infty) = 0. \end{cases} \tag{19}$$

Given a boundary condition $f(x) \in L^1(\mathbb{R})$, the solution is given by the convolution of $f$ with the Poisson kernel $P(x, y) = \frac{1}{\pi} \frac{y}{x^2 + y^2}$ [48, 50], i.e., $\alpha(x, y) = \int_{-\infty}^{\infty} dz\, P(x - z, y) f(z)$, analogous to the discrete case. In particular, we want to study the decay of a solution localized around $(0, 0)$. First, let us take $f(x) = \delta(x)$, the Dirac delta distribution. This leads to a diverging solution at the boundary. Nevertheless, our goal is to understand the long distance behavior where the solution is well-behaved. Alternatively, we could use a regularized version of the delta distribution instead. In any case, this boundary condition gives $\alpha(x, y) = P(x, y)$, which decays as $\alpha \sim 1/r$ at large distances. Recall that $r$ refers to the orthogonal distance to the boundary. Hence, while it decays towards the bulk, the decay is too slow for the norm $\int dx\, dy\, |\alpha(x, y)|^2$ to converge.

Can we construct more localized solutions? Note that the above solution is reminiscent to the electrostatic potential generated by a point-like source at the boundary. This suggests a natural way of constructing solutions with a faster decay: replace the point-like source with a dipole or some higher multipole source. This can be achieved by using a boundary condition that is a higher derivative of the delta distribution: $f(x) = \delta^{(n)}(x) \equiv \partial_x^n \delta(x)$. Using this boundary condition one finds, after integrating by parts, the solution $\alpha^{(n)}(x, y) = (-1)^n \partial_z^n P(x - z, y)|_{z=0}$. These decay asymptotically as $\alpha^{(n)}(0, y) \sim y^{-(n+1)}$, making them increasingly localized as we make $n$ larger. In particular, for any $n \geq 1$ they decay sufficiently quickly to make their norm convergent.

We now want to find an analogous set of solutions on the lattice. To do so, we first need to discretize the boundary condition $f(x) = \delta^{(n)}(x)$. This can be simply done by replacing the derivatives by (central) finite differences of the Kronecker delta $\delta_{i,0}$. In particular, the $n$-th derivative of a lattice function $f_i$ at site $i$ is given by $\Delta_i^n f_i = \sum_{k=0}^n \binom{n}{k}(-1)^k f_{i+\frac{n}{2}-k}$, (here and below we focus on $n$ even). Hence, for $f_i = \delta_{i,0}$, this requires fixing $n + 1$ non-zero boundary

---

[7] Note the change of notation: in this section $(0, 0)$ refers to a location in the middle of the boundary.

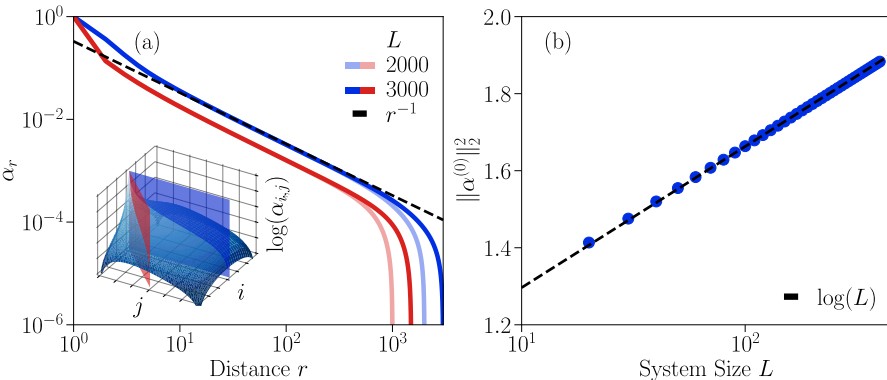

Figure 7: **Localization of $n = 0$ charges.** (a) Decay of $\alpha_{i,j}^{(0,\frac{L}{2})}$ towards the bulk along two different directions specified by the two planes in the 3D plot. In both cases, the solution decays as a $1/r$ with the distance $r$. (b) Divergence of the 2-norm of $\alpha_{i,j}^{(0,\frac{L}{2})}$ in the limit $L \to \infty$.

values given by

$$\alpha_{i,0}^{(n),\mathrm{B}} = (-1)^i \binom{n}{\frac{n}{2} - i} \Big/ \binom{n}{\frac{n}{2}} , \tag{20}$$

for $i \in \{-\frac{n}{2}, \dots, 0, \dots, \frac{n}{2}\}$. We have normalized $\alpha^{(n),\mathrm{B}}$ such that $\alpha_{0,0}^{(n),\mathrm{B}} = 1$ for all $n$. Given this boundary condition, we can find the solution of Eq. (8) in the bulk. In Fig. (6) we plot the solution for $n = 2$ and $n = 4$. The two different colors emphasize the change of sign of the solution along the nodal lines where $\alpha^{(n)}$ vanishes. E.g., $\alpha^{(2)}$ is positive in the central lobe (red) and negative on the sides (blue). A higher $n$, gives $n$ nodal lines with the corresponding changes of sign. The solutions $\alpha^{(n)}$ give rise to a set of conserved quantities which we denote by $\mathcal{Q}_{\mathbf{r}_0}^{(n)}$, where $\mathbf{r}_0$ refers to the location around which they are localized ($\mathbf{r}_0 = (0,0)$ in the discussion above).

The solutions $\alpha^{(n)}$ inherit their properties from their continuum counterparts. For $n = 0$, where the boundary condition is simply a Kronecker delta, we find a $1/r$ decay in $\alpha_{i,j}^{(0)}$, as shown in Fig. 7a. This means that the norm of $\mathcal{Q}^{(0)}$ diverges logarithmically with the linear size of the system, as shown in Fig. 7b. Hence, these charges are not sufficiently strongly localized at the boundary for a single one of them to give a finite Mazur's bound. Instead, we can make use of the general expression for Mazur's bound (14) and include all of them simultaneously. We find that this is sufficient to obtain a lower bound that is tight to the result obtained in our numerical simulations for any $S$ (see dashed black line in Fig. 1b). Additional details can be found in Appendix A and Ref. [46].

On the other hand, in agreement with the continuum case, the solutions for higher $n$ decay faster, as $r^{-(n+1)}$, as we confirm numerically in Fig. 8a. As a consequence, a *single* one of these charges is sufficient to give a finite Mazur's bound, using Eq. (15). However, while the lattice and continuum problems match at long distances, there are differences in their behavior close to the boundary that affect Mazur's bound. In particular, while the charges become more strongly localized towards the bulk for higher $n$, they also become more spread out at the boundary, which leads to an increase in their 2-norm. As a consequence, the value of the bound (15) in fact *decreases* with $n$ (while remaining finite for any finite $n$). Let us provide a rough estimate. For sufficiently large $n$, $\alpha^{(n)}$ decays quickly away from the boundary. Motivated by this, we estimate the scaling of $\|\alpha^{(n)}\|_2^2$ with $n$, by replacing it with the norm of the contribution from the boundary alone, $\|\alpha^{(n)}\|_2^2 \sim \|\alpha^{(n),\mathrm{B}}\|_2^2$. The latter can be evaluated from Eq. (20) and gives $\|\alpha^{(n),\mathrm{B}}\|_2^2 \sim \sqrt{n}$ in the limit of large $n$. We evaluate Mazur's bound

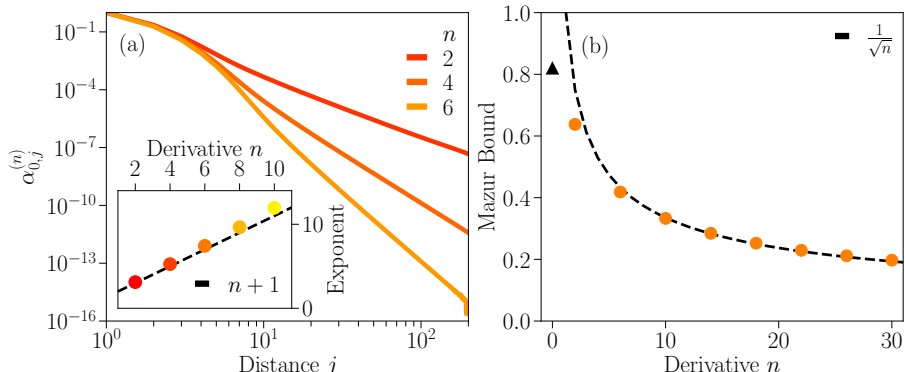

Figure 8: **Mazur's bound and higher-moment distributions.** (a) Power-law decay of higher-moment charges with modulation $\alpha_{0,j}^{(n)} \sim j^{-(n+1)}$ as predicted from the continuum Laplace equation. (b) Dependence of Mazur's bound (15) with $n$, when including a single higher-moment charge with modulation $\alpha^{(n)}$. Mazur's bound including all $n = 0$ charges at the boundaries is shown as a black triangle. For this data we used a linear system size of $L = 300$.

exactly for various $n$ in Fig. 8b, and find that it agrees with this estimate. The same figure also shows Eq. (14) evaluated using *all* $n = 0$ charges taken together, which gives a stronger bound. In Appendix A we show the finite-size scaling of Mazur's bound with system size for several $n$.

All together, we managed to construct different families of conserved quantities $\{\mathcal{Q}_{\mathbf{r}_0}^{(n)}\}$, which become more and more localized towards the bulk when considering higher and higher finite differences. Different $\mathcal{Q}_{\mathbf{r}_0}^{(n)}$ are localized around a boundary site with coordinates $\mathbf{r}_0$. For a given $n$, $\alpha^{(n),\text{B}}$ vanishes everywhere except on $n + 1$ sites centered around $\mathbf{r}_0$. The set of charges where $\mathbf{r}_0$ are at distance $\frac{n}{2}$ along the boundary are then trivially linearly independent. Hence, this proves that there exist at least $\mathcal{O}(L)$ of those. Nevertheless, different families parametrized by different $n$'s are not linearly independent of each other. In particular, we can use the $n = 0$ charges to construct all other families.

## 4.3 Generalizations

So far we have focused on the particular square lattice model defined in Eq. (1). However, our construction can be extended to higher dimensions as well as to different lattices as long as the associated recurrence relation

$$n_0 \alpha_{\mathbf{r}} + \sum_{\mathbf{v} \in \mathcal{N}_{\mathbf{r}}} n_{\mathbf{v}} \alpha_{\mathbf{v}} = 0, \tag{21}$$

corresponds to a discrete Laplace equation, i.e., its solutions are discrete harmonic functions. Some examples of models satisfying this requirement are shown in Fig. 1. For example, while the models in Figs. 1(a,b) are defined on different lattices, they both correspond to different discretizations of the same continuum Laplace equation. Hence the same long-distance decay for conserved quantities localized at the boundary of the system applies to them.

To extend the construction to higher dimensions, we now consider the continuum Laplace equation in $d+1$ dimensions in the hyperplane $\mathbb{H}^{d+1} = \{(\mathbf{x}, y) \in \mathbb{R}^{d+1} | y > 0\}$. The corresponding Poisson kernel is a generalization of the 2D one and reads $P_d(\mathbf{x}, y) = c_d y(y^2 + \|\mathbf{x}\|^2)^{-\frac{d+1}{2}}$, with some dimension-dependent constant $c_d$ [50]. Once again the squared 2-norm of the (regularized) solutions with $\alpha^{\text{B}}(\mathbf{x}, 0) = \delta(\mathbf{x})$ diverges logarithmically in the thermodynamic limit

and hence these are not sufficient to prove finite boundary correlations. Instead we can again consider "multipole" boundary conditions. Fixing $\alpha^{\text{B}}(\mathbf{x}, 0) = \delta^{(\mathbf{n}_d)}(\mathbf{x}) \equiv \partial_{x_1}^{n_1} \dots \partial_{x_d}^{n_d} \delta(\mathbf{x})$, the decay of the solution at large distances is given by $\alpha^{(\mathbf{n}_d)}(\mathbf{0}, y) \propto y^{-\sum_{i=1}^{d} n_i - d}$. Therefore, we find that one can construct charges that are sufficiently localized at the $d$-dimensional boundaries of the system as to provide a finite Mazur's bound.

The previous discussion appears to suggest that as long as the continuum limit of the linear recurrence Eq. (21) is given by the Laplace equation, one should be able to show that the boundary correlations are finite. However, some subtleties need to be addressed. First of all, recall that we explicitly made use of the fact that solutions of Eq. (21) are discrete harmonic functions. This ensured that for any boundary condition (and any system size) there exist a unique solution. If this was not the case, even the existence of a solution is not ensured. One such example is the 2D model defined in Eq. (5) of Ref. [37]. Our construction of boundary charged does not apply in that case; however, we have observed numerically that boundary correlations nevertheless fail to decay even in that model.

One set of models to which our previous discussion *does* directly apply are generated by the following local gates:

$$G_{(n_1, n_2, n_3, n_4)} = \{n_{-1,0}, n_{0,1}, n_{0,0}, n_{0,-1}, n_{1,0}\}_{\mathbf{r}} = \{n_1, n_2, -N, n_3, n_4\}_{\mathbf{r}}, \tag{22}$$

with $N = \sum_i n_i$. These correspond to the following recurrence relation,

$$\alpha_{i,j} = p_1 \alpha_{i+1,j} + p_2 \alpha_{i-1,j} + p_3 \alpha_{i,j+1} + p_4 \alpha_{i,j-1}, \tag{23}$$

with $p_i = n_i / N$. These equations were solved exactly in Ref. [47]. When $n_1 = n_2$ and $n_3 = n_4$, the continuum limit of this recurrence reads

$$\left( p \partial_x^2 + (1-p) \partial_y^2 \right) \alpha_{\text{in}}(x, y) = 0, \tag{24}$$

where we defined $p = 2p_1$. Solutions of this equation can be found by performing the change of variables $x \to x/\sqrt{p}$, $y \to y/\sqrt{1-p}$, and then are given by the evaluation of the solution of the isotropic problem (19) at $\alpha_{\text{in}}(x, y) = \alpha(x/\sqrt{p}, y/\sqrt{1-p})$. Hence, our discussion about the localization properties of boundary charges also applies to these family of anisotropic models.

# 5 Quantum systems and strong zero modes

So far our discussion has focused on classical Markov chain dynamics. However, it is easy to map the systems we studied onto quantum Hamiltonians that share the same symmetries. Given a set of local gates $G_{\mathbf{r}} = \{n_{\mathbf{v}}\}_{\mathbf{v} \in \mathcal{R}_{\mathbf{r}}}$, acting on some region $\mathcal{R}_{\mathbf{r}} = \{\mathbf{r}\} \cup \mathcal{N}_{\mathbf{r}}$ centered on a site $\mathbf{r}$, one can construct a quantum spin-$S$ Hamiltonian $\hat{H}_{\text{G}} = \sum_{\mathbf{r}} J_{\mathbf{r}} \hat{h}_{\mathbf{r}}$, on the same lattice as

$$\hat{h}_{\mathbf{r}} = \left( \hat{S}_{\mathbf{r}}^{\text{sgn}(n_0)} \right)^{|n_0|} \bigotimes_{\mathbf{v} \in \mathcal{N}_{\mathbf{r}}} \left( \hat{S}_{\mathbf{v}}^{\text{sgn}(n_{\mathbf{v}})} \right)^{|n_{\mathbf{v}}|} + \text{h. c.}, \tag{25}$$

where $\text{sgn}(n_{\mathbf{v}})$ is the sign of $n_{\mathbf{v}}$, and $J_{\mathbf{r}}$ is an arbitrary choice of real coefficients. By construction, when written in the computational basis, $\hat{H}_{\text{G}}$ has the same block-diagonal structure as the original Markov generator built from the gates $G_{\mathbf{r}}$. In particular, it shares both its fragmentation properties and its spatially modulated symmetries (with $\sum_{\mathbf{r}} \alpha_{\mathbf{r}} s_{\mathbf{r}}$ replaced by $\sum_{\mathbf{r}} \alpha_{\mathbf{r}} \hat{S}_{\mathbf{r}}^z$). These properties are also preserved by any additional diagonal terms $\hat{V}(\{\hat{S}_{\mathbf{r}}^z\})$ that are function of the $\hat{S}_{\mathbf{r}}^z$ only.

However, $\hat{H}_G$ might also have additional symmetries, not present in the classical model. Of particular interest are $\mathbb{Z}_2$ symmetries like $R_x = \prod_{\mathbf{r}} e^{i\pi \hat{S}_{\mathbf{r}}^x}$ and $R_y = \prod_j e^{i\pi \hat{S}_j^y}$. Indeed, $R_x \hat{S}_{\mathbf{r}}^{\pm} R_x = \hat{S}_{\mathbf{r}}^{\mp}$ and $R_x \hat{S}_{\mathbf{r}}^z R_x = -\hat{S}_{\mathbf{r}}^z$ so the Hamiltonians constructed above are invariant under $R_x$ as long as $\hat{V}$ is built up from even products in the local $\hat{S}^z$ operators. The importance of these additional discrete symmetries stems from the fact that they anticommute with the spatially modulated symmetries and therefore lead to exact degeneracies in the many-body spectrum [38].

To have these degeneracies, any spatially modulated symmetry of the form $\sum_{\mathbf{r}} \alpha_{\mathbf{r}} S_{\mathbf{r}}^z$ is sufficient, along with one of the aforementioned $\mathbb{Z}_2$ symmetries. For the discrete Laplacian models there are always such conserved quantities, independent of the choice of boundary conditions. However, we can modify the models in a way such that only the charges localized at the boundaries remain, which exist only for open boundaries. We do so by defining the local gates

$$G^{(p)} = \{n_{-1,0}, n_{0,1}, n_{0,0}, n_{0,-1}, n_{1,0}\}_{\mathbf{r}} = \{1, 1, -N, 1, 1\}_{\mathbf{r}}, \tag{26}$$

with associated recurrence relation reads

$$N\alpha_{i,j} = \alpha_{i+1,j} + \alpha_{i-1,j} + \alpha_{i,j+1} + \alpha_{i,j} + \alpha_{i,j-1}. \tag{27}$$

Choosing $N > 4$ rules out the conservation of the global charge and hence of any of its higher-moments. In general, solutions of this recurrence equation are not discrete harmonics, but rather correspond to the eigenvalue problem $\triangle \alpha = \varepsilon \alpha$ with $\varepsilon \neq 0$ in the continuum. In this case, one only finds solutions of the form $\alpha_{i,j} = (\xi_x)^i (\xi_y)^j$ and $\alpha_{i,j} = (\xi_x)^i e^{ik_y j}, e^{ik_x i} (\xi_y)^j$; both localized near the boundary. Importantly, this family of models does not show spatially modulated global conserved quantities for periodic boundary conditions. We thus expect them to feature similar phenomenology to that of the strong zero modes (SZM) introduced in Ref. [38], with degeneracies throughout the many-body spectrum for open, but not for closed boundaries. This construction can be extended to higher dimensions by for example considering a $d$-dimensional cubic lattice with $n_{0,0} = -N$, where $N > 2d$, and the remaining contributions equals to $+1$ as in Eq. (26). These are higher-dimensional generalizations of the 1D models with exponentially localized symmetries introduced in Ref. [37].

Nevertheless, despite the similarities with SZM, some important differences remain. First, while the boundary modes of Ref. [38] are only approximately conserved for finite systems, ours are exact for any $L$. This somewhat changes the logic of the construction: Instead of using the zero mode to toggle between the two different symmetry sectors of the exact $\mathbb{Z}_2$ symmetry, we can also classify energy eigenstates using the boundary symmetries. Arguably, the most important difference to standard SZM is that our boundary modes correspond to continuous, rather than discrete symmetries (i.e., there is no "normalization condition" of the form $(\hat{\mathcal{Q}})^m = 1$ for any integer $m$). This condition appears to be "highly non-trivial and fundamental" [51] to ensure a non-zero radius of convergence in the perturbative construction of such modes (indeed, our boundary modes are presumably highly sensitive to any additional perturbations). While this condition appears to hold for previous constructions of SZM found in the literature, it is not clear whether it should be generally imposed [51]. We therefore leave it to future work to determine whether the models introduced here can be meaningfully fit into the framework of SZM.

# 6  Conclusions and outlook

In this work we studied a family of models, which we named discrete Laplacian models, and which can be defined on an arbitrary lattice (or, more generally, bounded-degree graph). We

proved two main results about these models. First, we proved that bulk auto-correlations saturate to a finite value when the on-site configuration spaces are chosen to take their minimal values. Our proof works by explicitly constructing spatial regions whose configurations are left unchanged by the dynamics, which can then be used to divide the rest of the system into disconnected regions.

Secondly, we constructed a hierarchy of linearly (in the linear system size) many conserved quantities, which are localized at the boundary of the system. These are new instances of spatially modulated symmetries whose modulations satisfy a discrete Laplace equation with Dirichlet boundary conditions. We showed that while these are only power-law (rather than exponentially) localized, the power of decay can be systemically increased by choosing appropriate boundary conditions. As a result, we are able to prove that boundary correlation are finite, by making use of Mazur's bound [49].

There are a number of questions left open by our work. One interesting aspect of our work is the construction of strongly fragmented models in higher dimensions, whereas previous examples in the recent literature were explicitly one-dimensional in nature, relying on the conservation of certain one-dimensional patterns due to hard-core constraints [25,36]. While we proved the presence of strong fragmentation in discrete Laplacian models, constructing the conserved quantities to label all connected components, and understanding their algebraic structure in the spirit of Refs. [25,36], is an interesting challenge.[8] This would shed more light on their behavior and would help in finding other strongly fragmented higher dimensional models.

While we proved strong fragmentation only in the case when we restricted the local spin (number of particles per site) to its smallest possible value, it is likely that even away from this limit, one would find a transition from a weakly to a strongly fragmented regime by tuning the density of particles per site, analogously to the one-dimensional case studied in Ref. [44] ( as well as other kinetically constrained models [35,52]). Understanding the higher dimensional versions of this transition is another interesting open problem.

The relative novelty of the new classes of modulated symmetries we uncovered also naturally leads to many open questions. The most pressing one is to understand their level of fine-tuning. In particular, what minimal mathematical structure is required to realize such symmetries, and to which extent it is sufficient for them to just be approximately conserved. Moreover, this work provides a stepping stone for a more ambitious goal: classifying all the possible types of spatially modulated symmetries that a physical system with local interactions can possess. While in 1D (with a finite number of modulated symmetries) it appears that only $\alpha_j = r^j$ with $r$ an algebraic number are allowed, the infinitely many number of conserved quantities appearing in higher dimensions permit richer modulations.

On a different note, the constructions of quantum models in Sec. 5 with conserved quantities that are either localized in one direction and plane-wave like in the other, or localized at the corners is reminiscent of topological phases of matter where low-energy modes localized at the boundaries appear. The conserved quantities associated to corners in particular resemble the corner modes of higher order topological modes, which have been previously linked to systems with multipole symmetries [53]. Whether these analogies can be pushed further is an interesting open problem.

---

[8]We point out that conserved quantities with a strictly bounded spatial support can be easily ruled out—at least for hypercubic lattices—by generalizing the argument from Appendix H of Ref. [26].

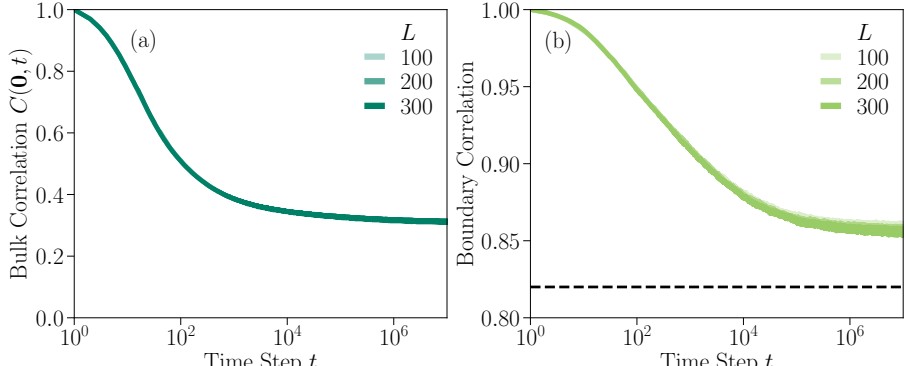

Figure 9: **Finite-size scaling of correlations.** (a) Finite-size scaling of bulk auto-correlations for $S = 2$ (a) and boundary auto-correlations (b) for model in Eq. (1). The data has been averaged over $N = 100$ and $N = 400$ circuit realizations respectively.

# Acknowledgements

We thank Juan Garrahan for bringing to our attention relevant references and for illuminating discussions about kinetically constrained models. We also thank Fabian Essler, Paul Fendley and Sanjay Moudgalya for insightful discussions. This research was financially supported by the European Research Council (ERC) under the European Union's Horizon 2020 research and innovation program under grant agreement No. 771537. F.P. acknowledges the support of the Deutsche Forschungsgemeinschaft (DFG, German Research Foundation) under Germany's Excellence Strategy EXC-2111-390814868. F.P.'s research is part of the Munich Quantum Valley, which is supported by the Bavarian state government with funds from the Hightech Agenda Bayern Plus. T.R. is supported in part by the Stanford Q-Farm Bloch Postdoctoral Fellowship in Quantum Science and Engineering.

# A  Finite-size Scaling Analysis

In this appendix we provide a more in-detail analysis of the finite-size scaling of the various numerical results we discussed in the main text.

Fig. 9a shows that the finite saturation value of bulk auto-correlations for the model in Eq. (1) when $S = 2$, does not scale down with system size, being the data converged in system size for all shown times. This data has been obtained for periodic boundary conditions and averaged over $N = 100$ circuit realizations. Fig. 9b provides the finite-size scaling of boundary auto-correlations. Here, we required $N = 400$ simulations to decrease the late-time fluctuations.

In the following we show the scaling of Mazur's bound with system size. We start with its scaling when only including a single higher-moment charge $\mathcal{Q}_{\mathbf{r}_0}^{(n)}$. This is shown in Fig. 10a for several $n = 2, 4, 6, 8, 10$. In addition, we also study the finite-size scaling when including $\mathcal{O}(L)$ different $n = 0$ charges Fig. 10b. We first note, that while linearly independent, these conserved quantities are not orthogonal with respect to the infinite temperature inner product

$$2\langle \mathcal{Q}_{\mathbf{r}_0}, \mathcal{Q}_{\mathbf{r}_0'} \rangle = \frac{S(S+1)}{3} \sum_{i,j} \alpha_{i,j}^{\mathbf{r}_0} \alpha_{i,j}^{\mathbf{r}_0'} , \tag{A.1}$$

and thus, to compute $M_{s_{x,y}}$ for spin correlations $\langle s_{x,y}(t)s_{x,y}(0)\rangle$ at the boundary, we need to

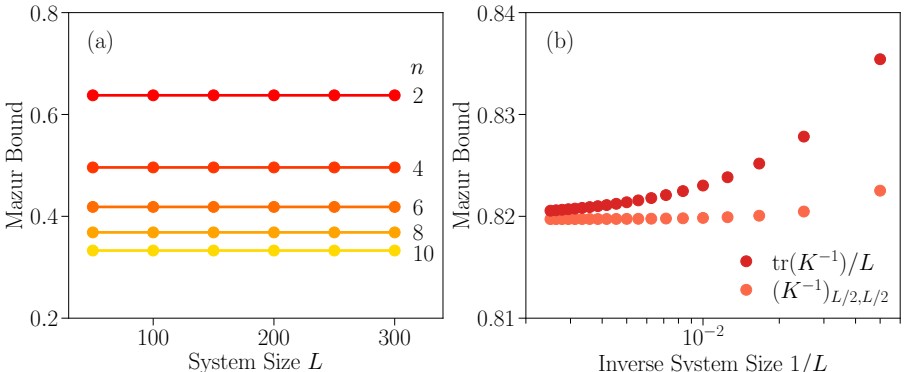

Figure 10: **Finite-size scaling of Mazur's bound.** Finite-size scaling of Mazur's bound (a) when including a single charge with modulation $\alpha^{(n)}$, and (b) when including $\mathcal{O}(L)$ charges with $n = 0$.

use the general expression (14). Without loss of generality and due to the $\mathbb{Z}_4$ symmetry of the lattice and local gates, we can restrict ourselves to focus solely on one boundary, e.g. on a site with coordinates $(0, y)$ where $1 \leq y \leq L$. As it turns out, we do not need to include all conserved quantities but only the ones of the form $\alpha^{\mathrm{B},a}_{i,j} = \delta_{i,0}\delta_{j,a}$ for $1 \leq a \leq L$, as all the remaining boundaries have exponentially small overlap with $s_{x,y}$. Consequently, one finds

$$\left\langle s_{x,y}, \mathcal{Q}_a \right\rangle = \frac{S(S+1)}{3}\alpha^{\mathrm{B},a}_{x,y} = \frac{S(S+1)}{3}\delta_{j,a}, \tag{A.2}$$

leading to

$$M_{s_{0,y}} = \frac{S(S+1)}{3}\sum_{a,b}\delta_{y,a}\cdot(K^{-1})_{a,b}\cdot\delta_{y,b} = \frac{S(S+1)}{3}(K^{-1})_{y,y}. \tag{A.3}$$

In the limit $L \to \infty$, this can give a finite value depending on the scaling of the diagonal matrix elements $(K^{-1})_{y,y}$ with $L$. Deriving the scaling of a particular matrix element $(K^{-1})_{y,y}$ with system size, however, is an analytically difficult task. We provide a finite-size scaling in Fig. 10b (orange dots). Alternatively, we can instead look for a lower bound of the averaged boundary auto-correlations

$$\lim_{T\to\infty}\frac{1}{T}\int_0^T \mathrm{d}t\,\frac{1}{4L}\sum_{\mathbf{r}\in B}\langle s_{\mathbf{r}}(t)s_{\mathbf{r}}(0)\rangle \geq M_{\mathrm{B}}, \tag{A.4}$$

where Mazur's bound can be expressed in terms of the trace of the inverse matrix $K^{-1}$

$$M_{\mathrm{B}} = \frac{S(S+1)}{3}\frac{\mathrm{tr}(K^{-1})}{L}. \tag{A.5}$$

This allows us to limit our calculations to a more general property of the matrix $K$. The scaling with system size is shown in Fig. 10b.

## B Counting of frozen states

To estimate the total number of frozen states of $G_+$ using Pauling's method, we consider the set of 5 sites on which a single gate acts and calculate the probability $P_{\mathrm{F}}$ that the configuration cannot be changed by the effect of the gate. To do so, we first compute the probability

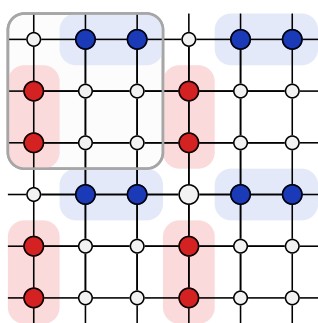

Figure 11: **Frozen lattice tiling.** Tiling defined by the unit cell (grey box) such that the spin configuration on the entire lattice is frozen for all times. Blue sites are highest, red sites lowest charge. All sites are frozen for any choice of $s_{\mathbf{r}} \in \{-S, \ldots, S\}$ on the white sites.

that the gate can fire (or anti-fire) at the beginning and then take the complement. Given our five degrees of freedom $s \in \{-2, -1, 0, 1, 2\}$ placed on a +-shape, we immediately see that in order for the gate to fire the central site has to be either $\pm 2$ and the outer sites are restricted to $\mp\{-1, 0, 1, 2\}$. With the total $5^5$ possible configurations, we have a "firing" probability $2 \cdot 1^1 \cdot 4^4/5^5 = 2^9/5^5$. Taking the complement of the probability gives $P_{\mathrm{F}} = 1 - \frac{2^9}{5^5} \approx 84\%$, so more than 80% of all initial +-shaped configurations of five charges are frozen. Say we now have a square lattice with $L^2$ sites. There are in total $(L-2)^2$ +-shaped configurations in the lattice all of which have the same probability $P_{\mathrm{F}}$ to be frozen initially. Neglecting the effects of overlapping configurations, the probability that the whole lattice is initially frozen and thus frozen for all times is simply given by $P_{\mathrm{F}}^{(L-2)^2}$. Therefore, the Pauling estimate for the total number of frozen states takes the form $N_{\mathrm{F}} = 5^{L^2}\left(1 - \frac{2^9}{5^5}\right)^{(L-2)^2}$.

# C  Numerical solution of linear recurrence relations

Finding a close-form solution of linear recurrence relations in terms of (arbitrary) boundary conditions is in general difficult. Hence, given the wide variety of recurrence relations we study in this work, we instead follow a numerical approach based on *Iterative Stencil Loops* [54, 55]. We then compare the results to those analytically obtained in the continuum limit. As in the case of $G_{\boxplus}$, we can arrange the general recurrence relation

$$n_0 \alpha_{\mathbf{r}} + \sum_{\mathbf{v} \in \mathcal{N}_{\mathbf{r}}} n_{\mathbf{v}} \alpha_{\mathbf{v}} = 0, \tag{C.1}$$

to yield an "averaging"-type recurrence relation:

$$\alpha_{\mathbf{r}} = -\frac{1}{n_0} \sum_{\mathbf{v} \in \mathcal{N}_{\mathbf{r}}} n_{\mathbf{v}} \alpha_{\mathbf{v}}. \tag{C.2}$$

A route for solving the latter is recursively constructing a solution by obtaining the value at a site $\mathbf{r}$ as given by the RHS of Eq. (C.2). To achieve this, we (1) start with an empty lattice initialized with the imposed boundary conditions $\alpha_{\mathbf{r}}^{\mathrm{B}}$ for $\mathbf{r} \in B$, (2) take the right sum as our "stencil" [54] and (3) iterating over the lattice updating every $\alpha_{\mathbf{r}}$ in the bulk with the respective weighted sum of its neighbors, in the sense that

$$\alpha_{\mathbf{r}}^{(t)} \longrightarrow \alpha_{\mathbf{r}}^{(t+1)} = -\frac{1}{n_0} \sum_{\mathbf{v} \in \mathcal{N}_{\mathbf{r}}} n_{\mathbf{v}} \alpha_{\mathbf{v}}^{(t)}, \tag{C.3}$$

where $\alpha_{\mathbf{r}}^{(t)}$ and $\alpha_{\mathbf{r}}^{(t+1)}$ are the values at the site before and after update, respectively. In particular for a lattice of size $|\mathcal{L}|$, this results to each iteration step taking $\mathcal{O}(|\mathcal{L}|^2)$ applications of the stencil. To terminate the algorithm, we compute the difference between two consecutive iteration steps $\mathcal{L}_t$ and $\mathcal{L}_{t+1}$ and stop the algorithm whenever a certain threshold is reached $\sum_{\mathbf{r}\in D}|\alpha_{\mathbf{r}}^{(t+1)} - \alpha_{\mathbf{r}}^{(t)}| < \varepsilon$. In particular, we choose $\varepsilon < 10^{-10}$. In the case of discrete harmonic functions, the algorithm is known to converge to the unique solution of the discrete Dirichlet's problem.

## D Solution of Eq. (9) via separation of variables

Analogously to the continuum case, we can solve Eq. (9) via separation of variables, i.e., $\alpha_{i,j} = X_i Y_j$. Let us explicitly follow the main procedure which will teach us something about the structure of the solutions. Using this ansatz the recurrence relation becomes

$$4X_i Y_j = (X_{i+1} + X_{i-1})Y_j + X_i (Y_{j+1} + Y_{j-1}). \tag{D.1}$$

After dividing by $X_i Y_j$ (assuming $\alpha_{i,j}$ does not vanish in the bulk) we find

$$4 = \frac{X_{i+1} + X_{i-1}}{X_i} + \frac{Y_{j+1} + Y_{j-1}}{Y_j}. \tag{D.2}$$

This implies that each term on the right hand side is a constant function of its argument and hence, they satisfy the one-dimensional recurrence relations

$$\begin{cases} X_{i+1} - 2X_i + X_{i-1} = \lambda X_i, \\ Y_{j+1} - 2Y_j + Y_{j-1} = -\lambda Y_j, \end{cases} \tag{D.3}$$

along the horizontal and vertical lattice directions respectively, for certain values of $\lambda \in \mathbb{R}$ which are fixed by the boundary conditions. To proceed we notice that being Eq. (8) linear, the Dirichlet problem can be solved adding up the solutions of four different Dirichlet problems with vanishing boundaries conditions except at a given boundary. This means that a general solution can be written as

$$\alpha_{i,j} = \alpha_{i,j}^{(1)} + \alpha_{i,j}^{(2)} + \alpha_{i,j}^{(3)} + \alpha_{i,j}^{(4)}, \tag{D.4}$$

with the different contributions solving the boundary problems

$$\begin{cases} Y_0 = Y_{N+1} = 0 \quad \text{and} \quad X_0 = 0, \quad \text{with} \quad \alpha_{N+1,j}^{(1)} = \chi_2(j), \quad \text{(I)} \\ Y_0 = Y_{N+1} = 0 \quad \text{and} \quad X_{N+1} = 0, \quad \text{with} \quad \alpha_{0,j}^{(2)} = \chi_1(j), \quad \text{(II)} \\ X_0 = X_{N+1} = 0 \quad \text{and} \quad Y_0 = 0, \quad \text{with} \quad \alpha_{i,N+1}^{(3)} = \eta_1(j), \quad \text{(III)} \\ Y_0 = Y_{N+1} = 0 \quad \text{and} \quad Y_{N+1} = 0, \quad \text{with} \quad \alpha_{i,0}^{(4)} = \eta_2(j). \quad \text{(IV)} \end{cases} \tag{D.5}$$

For example, solving problem (I) leads to $Y_{n,j} = \sin(k_y^n j)$ and $X_{n,i} = \sinh(\kappa_x^n i)$ with $k_y^n = n\pi/(N+1)$ for $n = 0, \ldots, N+1$. This also restricts the values of $0 < \lambda < 4$ to those satisfying $\cos(k_x^n) = 1 - \lambda_n/2$, and in turn $\cosh(\kappa_x^n) = 1 + \lambda_n/2$. Hence, we have found the fundamental solutions $A_n \sinh(\kappa_x^n i) \sin(k_y^n j)$, which by linear superposition lead to the general solution of problem (I)

$$\alpha_{i,j}^{(1)} = \sum_{n=0}^{N+1} A_n \sinh(\kappa_x^n i) \sin(k_y^n j), \tag{D.6}$$

where the coefficients $A_n$ are fixed by

$$\chi_2(j) = \sum_{n=0}^{N+1} A_n \sinh(\kappa_x^n(N+1)) \sin(k_y^n j),\tag{D.7}$$

i.e., proportional to the Fourier coefficients of $\chi_2(j)$.

Solving the three remaining problems, which again involve products of sinusoidal and hyperbolic functions, one can find a general solution of Eq.(9) for any choice of boundary functions. This implies that the fundamental solutions $\sinh(\kappa_x^n i)\sin(k_y^n j)$, $\sinh(\kappa_x^n[i-(N+1)])\sin(k_y^n j)$, $\sin(k_x^n i)\sin(\kappa_y^n j)$ , $\sin(k_x^n i)\sinh(\kappa_y^n[j-(N+1)])$ with $n = 0 \cdots, N+1$ form a basis for solutions of Eq. (9), showing that the number of independent symmetries scales with the linear system size. This approach not only gives us the fundamental solutions from where to obtain any other one, but also allows us to explicitly find particular spatial modulations that connect to the quasi-periodic and exponential modulations we encountered in the previous chapter. In general, we can solve Eqs. (D.3) finding the roots of the associated characteristic polynomials $r^2 - (2 \pm \lambda)r + 1 = 0$ which take the values

$$\begin{cases} x_{1,2} = \frac{2+\lambda}{2} \pm \frac{\sqrt{\lambda(\lambda+4)}}{2}, \\ y_{1,2} = \frac{2-\lambda}{2} \mp \frac{\sqrt{\lambda(\lambda-4)}}{2}, \end{cases}\tag{D.8}$$

for the first and second equations respectively. As the characteristic polynomial are palindromic, the two roots are inverse of each other. Whether these are real ($\eta_{x,y}$) or pure complex phases ($e^{ik_{x,y}}$) is determined by the sign of the discriminants $\Delta_x = \lambda(\lambda+4), \Delta_y = \lambda(\lambda-4)$: If $\Delta_{x,y} > 0$, the corresponding solution can be written in terms of hyperbolic or exponential functions; while $\Delta_{x,y} < 0$ correspond to sinusoidal or complex exponential modulations.

We can split the solutions into three main types: (i) $\lambda = 0$ contains all multipole conserved quantities we already identified except for the $x^2 - y^2$ quadratic moment. As this solution does not factorize in horizontal and vertical directions, but rather as $\alpha_{x,y} = (x+y)(x-y)$, we can only recover it by superposing many fundamental solutions. Alternatively, these could be explicitly found when writing the recurrence relation in terms of center of mass $s = i + j$ and relative $r = i-j$ coordinates, i.e., $\alpha_{i,j} = X_s Y_r$. The second case (ii) corresponds to exponential (hyperbolic) solutions, which are localized near the corners of the 2D lattice when $|\lambda| > 4$. These can be directly found solving Eq.(8) with the ansatz $\alpha_{i,j} = (\xi_x)^i (\xi_y)^j$. Finally, the third case (iii) corresponds to the product of sinusoidal and hyperbolic solutions along orthogonal directions, like e.g., $\alpha_{i,j} = (\xi_x)^i e^{-ik_y j}$. Thus, while solutions with a finite momentum mode along one of the lattice directions exist (for $|\lambda| < 4$), these are exponentially damped along the orthogonal direction and do not contribute to spin correlations in the bulk. Moreover, we can also rule out solutions of the form $\alpha_{i,j} \sim e^{ik_x i} e^{ik_y j}$ which would have led to conserved finite modes in the bulk correlations.

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
