# Peer review of "Fragmentation-induced localization and boundary charges in dimensions two and above"

_SciPost Physics, doi:SciPost Phys. 14, 140 (2023)_

## Round 1 · Referee Report · Alexey Khudorozhkov (Referee 1) · 2022-10-31

Strengths

Very clear paper that demonstrates the first (to my knowledge) example of strong Hilbert space fragmentation in 2 and higher dimensions, which the authors realize in a class of models defined on generic graphs.

A rigorous analytical proof of strong fragmentation is presented and supported by both numerical and analytical calculations of the correlation function. Localized conserved quantities and correlation functions at the boundary are studied in detail.

Weaknesses

The authors do not show that the class of models they introduce is non-integrable.

Report

This is a very solid, understandable and novel paper in the field of non-thermalizing systems and in particular in the subfield of Hilbert space (or, in the classical case, configuration space) fragmentation. It presents a solid demonstration that the strong Hilbert space fragmentation can also exist in higher than one dimension. Additional beauty lies in the fact that the class of models introduced in the paper is relatively simple, but nevertheless quite generic with respect to the considered lattice.

My main issue is the following. The main result of the paper would only be rendered interesting if the considered models are non-integrable. Otherwise, the absence of thermalization can be explained by emergent local conserved quantities (in the bulk). The authors neither provide any proof that local conserved quantities are absent in an infinite system, nor do they prove non-integrability numerically by e.g. studying the level spacing statistics. I believe that the absence of local conserved quantities might be shown analogously to Appendix H of https://arxiv.org/abs/2107.09690, generalized to arbitrary graphs and higher spins, although special care should be taken when dealing with higher-spin operators.

Below I also composed a list of minor issues/typos/comments/suggestions: 1. In Fig.2(a), it is not obvious that the curve for $S=2$ obeys $\sim t^{-1/2}$ behavior. I would recommend adding the corresponding line to the log-log plot. 2. How are boundary correlations defined? For example, in Fig. 2(b), it is not clear what $x$ stands for. Is it the direction along the boundary? I would recommend adding a brief explanation in the captions. 3. The inset of Fig.2(b) does not have a curve for $t=10^6$, although from other plots one can see that the authors have access to such times. Why so? 4. Page 7 below Lemma 1 contains several typos (please double check me on that): - In the line that starts with "In the situation above...", $F_v (t-1, t'+1)$ should be replaced with $F_v (t+1, t'-1)$. - In the line that starts with "Note that between the two firings...", $\Delta m_v (t, t') \equiv m_v(t) - m_v(t'+1) = z_v$ should be replaced with $\Delta m_v (t, t') \equiv m_v(t) - m_v(t') = z_v$. - In the unnumbered equation, $-4F_v(t,t')$ should be replaced with $-z_v F_v(t,t')$. - In the sentence "Between these two times...", $v'$ should be replaced with $v_1$. - In the sentence "We thus end up with an infinite regress...", the word "one" is missing in "but then (one) of its neighbors has to fire twice...". 5. In the statement of Corollary 1, I believe that $\bar{\mathcal{R}}\equiv \bigcup_{v\in\mathcal{R}} \mathcal{N}_v$ should be replaced with $\bar{\mathcal{R}}\equiv \mathcal{R} \cup \left(\bigcup_{v\in\mathcal{R}} \mathcal{N}_v \right)$, since the union of all neighbors of $\mathcal{R}$ does not necessarily include $\mathcal{R}$. A counterexample is when $\mathcal{R}$ is a single site or a disjoint union of single sites. 6. In addition, I think that Eq.(18) should be rewritten as $n_\mathbf{r}\alpha_\mathbf{r} + \sum_{\mathbf{v} \in \mathcal{N}\mathbf{r}} n\mathbf{v} \alpha_{\mathbf{r} + \mathbf{v}} = 0$, since $\mathcal{N}_\mathbf{r}$ does not include site $\mathbf{r}$. I also encourage the authors to check other usages of $\mathcal{N}$ for possible typos. 7. In the first sentence of the proof of Theorem 1, $s_{v'}=0$ should be replaced with $m_{v'}=0$. 8. In the last paragraph of section 3, the authors discuss the scaling of the number of frozen states. Although the authors do not make any claims that the exponential scaling is connected to strong fragmentation, I think it is important (especially for the readers not familiar with the topic) to emphasize that an exponential scaling of frozen states does not(!) imply strong Hilbert space fragmentation. I would suggest adding a sentence about that. 9. I believe that there is a simple rigorous lower bound on the number of frozen states that is tighter than the one calculated by the authors. Any state with no empty or maximally occupied sites is frozen. The number of such states is $(2S-1)^{L^2}$, which is larger than $(2S+1)^{\frac{5}{9}L^2}$ for any $S \geq 2$. 10. In Eq.(22), $\mathcal{R}_\mathbf{v}$ should be replaced with $\mathcal{R}_\mathbf{r}$.

Requested changes

  • Provide an analytical or numerical proof that the strong fragmentation in the models of interest does not arise from emergent integrability, i.e., prove that there isn't an extensive number of local conserved quantities in the bulk.

  • Correct the minor issues where and if the authors see fit.

  • validity: top
  • significance: high
  • originality: high
  • clarity: top
  • formatting: perfect
  • grammar: perfect

Author:  Pablo Sala de Torres-Solanot  on 2023-01-15  [id 3238]

(in reply to Report 1 by Alexey Khudorozhkov on 2022-10-31)
Category:
answer to question

We thank the referee for the positive assessment of our work, and we very much appreciate their detailed report providing very valuable comments. We have implemented all of them in the updated version of the manuscript.

''My main issue is the following. The main result of the paper would only be rendered interesting if the considered models are non-integrable. Otherwise, the absence of thermalization can be explained by emergent local conserved quantities (in the bulk). The authors neither provide any proof that local conserved quantities are absent in an infinite system, nor do they prove non-integrability numerically by e.g. studying the level spacing statistics. I believe that the absence of local conserved quantities might be shown analogously to Appendix H of https://arxiv.org/abs/2107.09690, generalized to arbitrary graphs and higher spins, although special care should be taken when dealing with higher-spin operators. ''

We assume that by ''intgrable'', the referee means a model exhibiting a (possibly complete) set of strictly local conserved quantities, akin to e.g. commuting projector models; indeed, other notions of integrability, such as Bethe Ansatz, do not seem relevant, since we consider systems in arbitrary dimensions and can break translation invariance in ways that do not change any of our conclusions.

First, let us point out that in the one-dimensional version of the strongly fragmented models we study, a full set of conserved quantities labeling the fragmented sectors is known (see ArXiv 1904.04266 and PRB 101.125126). In that case, the conserved quantities are indeed non-local; instead they correspond to string-like operators. Moreover, they are not complete and allow for sectors of the Hilbert space that are exponentially large. It can be shown that at least some of these exhibit thermalizing dynamics within the sector. We expect similar conclusions to hold in the more general models we consider here.

Moreover, as the referee suggests, we can prove directly that our models do not have strictly local conserved quantities. In the following we prove that this is indeed the case for cubic lattices in any spatial dimension $d$, and for any spin representation. In this setup the arguments presented in Appendix H of SciPostPhys 13.4.098 (in the following cited as Ref. [1]) can be easily generalized. The proof works by contradiction. We assume that there exists a conserved local operator $Q$, with local and bounded support supp$(Q)$ contained within a bounded region $R_Q$. Here, supp$(Q)$ is defined as the set of sites where $Q$ acts non-trivially. This operator can be written as

$$ Q=\sum_{{\mu}}c_\mu\bigotimes_{\vec{r}\in\mathcal{R}Q}\lambda^{\mu, $$}}}_{\vec{r}
where $\mu\in {1,\dots,(2S+1)^2}$ and $\lambda$ are the generalized Gell-Mann matrices ---including the identity, $1$--- forming an orthogonal basis for the space of $(2S+1)\times (2S+1)$ complex matrices, and $c_\mu$ are arbitrary complex coefficients.

Let us represent all sites that lie inside $R_Q$ as having coordinates $\vec{r}=(x_1,x_2,\dots,x_d)$. W.l.o.g. pick the minimum value of the set of first components $x_{1,min}=\min{x_1}$ and only look at sites with $r_{face}=(x_{1,min},x_2,...,x_d)\in$ supp $(Q)$, i.e. one particular ''face'' of the region supp$(Q)$ traces. Utilizing the $+$-shaped pattern of the local regions our Hamiltonian acts upon via local terms $h_{\vec{r}}$, we can choose a particular site $\vec{r}^\prime=(x_{1,min},x^\prime_2,\dots ,x^\prime_d)\in$ supp$(Q)$ and centering $h_r$ at $(x_{1, min}-1,x^\prime_2,...,x^\prime_d)$ ; this is one site away from the boundary of supp$(Q)$. Hence, $h_{\vec{r}}$ only acts on a single site of supp$(Q)$. By the form of the operator and the fact that $[Q,H]=0$, for any non-zero $c_\mu$ we must have $\lambda^{\mu_{\vec{r}'}}_{\vec{r}'}=1$. However, this is a contradiction to $\vec{r}'$ belonging to the support of $Q$. Therefore, the initial assumption about the existence of a local conserved operator $Q$ is wrong. We added a footnote to the main text to point this out.

Minor issues/typos/comments/suggestions

  1. We have added a line highlighting the diffusive behavior mentioned in the text to Fig. 2(a).
  2. We have clarified the definition of the boundary correlation as appearing in the insert Fig. 2(b).
  3. As pointed out by the referee, the data points for the latest available time were absent from the insert Fig. 2(b). This was indeed a mistake in the legend displayed, which we corrected.
  4. We have fixed several minor mistakes in Lemma 1 and Theorem 1 including those pointed out by the referee:

    (i) $F_v (t-1,t'+1)$ has been replaced with $F_v (t+1,t'-1)$, in the line starting with ''In the situation above ...''. (ii) $\Delta m_v (t,t')\equiv m_v (t)-m_v (t'+1)=z_v$ in the line starting with `''Note that between the two firings ...'' now reads $\Delta m_v (t,t')\equiv m_v (t)-m_v (t')=z_v$. (iii) There was one unnumbered equation, which now has the correct numbering, and the prefactor of $F_v (t,t')$ was replaced from $4$ to $z_v$. (iv) $v'$ in the sentence ''Between these two times ...'' now correctly reads $v_1$. (v) In the sentence iiWe thus end up with an infinite regress ...'', the word ``one'' was missing, which is now added.

  5. Per the suggestion of the referee, we checked our use of the neighborhood set and corrected some inconsistencies thereof.

  6. We have replaced $s_{v^\prime}=0$ by $m_{v^\prime}=0$ in the first sentence of the proof of Theorem 1.
  7. We believe that the updated version of the manuscript, which now includes Appendices C and D in Section 3.3., makes clear the difference between strong fragmentation of the configuration space, and the presence of exponentially many frozen states, which can also appear in weakly fragmented systems. We do so by first showing that the systems we considered do showcase strong fragmentation, and only then discussing the scaling of the number of frozen states.
  8. We further want to thank the referee for pointing out a simple lower bound on the number of frozen states. While it is only valid for spin $2$, we have included it in the main text as well as in Fig.4.
  9. We have corrected and updated the notation regarding the use of $\vec{v}$ and $\vec{r}$ in several parts of the text.

Attachment:

report_2.pdf

---

## Round 1 · Referee Report · Anonymous (Referee 2) · 2022-11-9

Report

This paper analyzes a classical stochastic model with a strong form of fragmentation. The main example is in two dimensions, but the model and the behavior do generalize to higher dimensions. One nice feature is that they are able to prove (as in rigorously) that individual spins have a non-vanishing probability of "remembering" their initial state for an arbitrarily long time. Another nice feature is that the model can be extended so that the only long-term coherence arises on the boundary. The resulting conservied currents greatly resemble the "strong zero modes" appearing in some quantum spin chains. The authors make some interesting comments about the connection, pointing out that their constuction seems to be more general than the earlier.

I thus recommend that the paper be published in SciPost. It is written clearly, and the results are described in a precise fashion. They are interesting, and very much part of the fragmentation story of great current interest.

I have only one comment. The authors adhere to the current PRL-based fashion of putting important and/or illuminating results in appendices. I thus suggest that several of the appendices be incorporated in the main text. Appendix B contains a simple but important step in the proof of the most important result in the paper. It's only a paragraph long, why isn't it included in the rest of the proof? Appendix D contains some very useful intuition into the bound proved, and is rather simple too. I'd incorporate that too.
  • validity: -
  • significance: -
  • originality: -
  • clarity: -
  • formatting: -
  • grammar: -

Author:  Pablo Sala de Torres-Solanot  on 2023-01-15  [id 3237]

(in reply to Report 2 on 2022-11-09)

We thank the referee for their accurate summary of our work, and their positive assessment.

We very much appreciate the constructive suggestion made by the referee, which has improved the presentation of our results. Following their suggestion, we have updated the text to include: The lower bound on the number of frozen states applicable to every spin S, previously contained in Appendix B, in Section 3.2. Section 3.3, with the updated title ``Strong fragmentation of the configuration space'', now includes Appendices C and D regarding the argument about the strong fragmentation of the configuration space and the lower for frozen states respectively. The content of the former now appears as proof of Theorem 2.

---

## Round 2 · Referee Report · Alexey Khudorozhkov (Referee 1) · 2023-1-17

Report

The authors addressed the main raised issue. Indeed, the proof from Appendix H of SciPostPhys 13.4.098 straightforwardly generalizes to higher-spin models with the "discrete Laplacian" Hamiltonian on the hypercubic lattice (and perhaps on any translationally invariant graph). Now, I am convinced that such models do not exhibit any strictly local conserved quantities and that the strong fragmentation originates from the restricted dynamics of the model, rather than from the local integrals of motion.

I have another minor issue about the revised manuscript and a possible typo:

  1. The bound for the number of frozen states I gave in the previous review, $(2S-1)^{L^2}$, is valid not only for $S=2$ (as the authors claim in the revised manuscript), but for any $S$. What's more, for any $S\geq 2$, this bound is larger than the bound calculated from tiling the plane with specific configurations, $(2S+1)^{\frac{5}{9}L^2}$. Therefore, the claim "Nevertheless, we can also derive a less tight but rigorous lower bound which holds for all half-integer S." is a bit meaningless. The bound $(2S-1)^{L^2}$ is no less rigorous and is a better bound for any $S\geq 2$. The $(2S+1)^{\frac{5}{9}L^2}$ bound is only tighter for $S=1/2, 1, 3/2$.

  2. In the proof of Theorem 2: "If a configuration has n frozen sites, i.e. sites whose state cannot evolve, then it is connected at most to $M^{L−n}$ other configurations.". Did the authors perhaps mean $M^{L^d -n}$? Otherwise, it is not clear to me why it is $M^{L-n}$.

  • validity: -
  • significance: -
  • originality: -
  • clarity: -
  • formatting: -
  • grammar: -

Author:  Pablo Sala de Torres-Solanot  on 2023-02-21  [id 3387]

(in reply to Report 2 by Alexey Khudorozhkov on 2023-01-17)
Category:
reply to objection

We thank the referee for noticing the typo pointed out in the second comment. We have corrected it in the new version of the draft substituting $M^{L-n}$ by $M^{L^d-n}$.

We also thank the referee for the other valuable comment. However, we remain uncertain about some aspects of the bound. We do not understand why the referee claims the bound $(2S-1)^{L^2}$ as the argument they gave in their previous reply ''Any state with no empty or maximally occupied sites is frozen'' seems to apply only to $S=2$. In general, any local state with $m\in[0,3]$ cannot fire and any $m\in[2S-3,2S]$ cannot anti-fire. However, the intersection is non-trivial only for $S=2$ (giving rise to the bound suggested by the referee) as well as for $S=5/2$ and $S=3$. In fact, it is easy to find configurations that are not frozen and have no site with $m=0$ or $m=2S$: For example take $S=3$ (i.e, $m$ takes values in $\{0,1,\dots,5,6\}$) and make $m=4$ everywhere. Then any site can fire. Similarly for half-integer spins, e.g. $S=5/2$ taking values $\{0,1,\dots,4,5\}$, choosing $m=3$ everywhere has the same effect. This generalizes to any spins $S>2$ and as such, we do not see the argument for the proposed bound.

Alexey Khudorozhkov  on 2023-02-22  [id 3389]

(in reply to Pablo Sala de Torres-Solanot on 2023-02-21 [id 3387])

Sorry, I was wrong about the bound. You are right, it only applies to $S=2$.

---

## Round 2 · Referee Report · Anonymous (Referee 2) · 2023-1-17

Report

The authors made the pedagogical changes I suggested, and so I am happy to recommend publication.

---

## Round 2 · List of Changes

We have updated the text to include: The lower bound on the number of frozen states applicable to every spin $S$, previously contained in Appendix B, in Section 3.2. Section 3.3, with the updated title ``Strong fragmentation of the configuration space'', now includes Appendices C and D regarding the argument about the strong fragmentation of the configuration space and the lower for frozen states respectively. The content of the former now appears as proof of Theorem 2. We have also corrected several typos appearing in the text.

---

## Round 4 · List of Changes

- We have included a new sentence in the second paragraph of page 5: "We have also numerically confirmed that spatial correlations remain localized to a finite region."
- We have corrected "L" as "L^d" whenever necessary in the statement and proof of Theorem 2 in page 10.
- We have eliminated previous appendix B "Strong fragmentation of the configuration space" which had the same content as the current proof of Theorem 2.

---

## Editorial Decision

published